# Let Me Think! A Long Chain-of-Thought Can Be Worth Exponentially Many Short Ones

**Parsa Mirtaheri**[*]
UC San Diego
parsa@ucsd.edu

**Ezra Edelman**[*]
University of Pennsylvania
ezrae@seas.upenn.edu

**Samy Jelassi**
Harvard University
sjelassi@fas.harvard.edu

**Eran Malach**
Harvard University[†]
emalach@g.harvard.edu

**Enric Boix-Adserà**
University of Pennsylvania
eboix@wharton.upenn.edu

## Abstract

Inference-time computation has emerged as a promising scaling axis for improving large language model reasoning. However, despite yielding impressive performance, the optimal allocation of inference-time computation remains poorly understood. A central question is whether to prioritize sequential scaling (e.g., longer chains of thought) or parallel scaling (e.g., majority voting across multiple short chains of thought). In this work, we seek to illuminate the landscape of test-time scaling by demonstrating the existence of reasoning settings where sequential scaling offers an exponential advantage over parallel scaling. These settings are based on graph connectivity problems in challenging distributions of graphs. We validate our theoretical findings with comprehensive experiments across a range of language models, including models trained from scratch for graph connectivity with different chain of thought strategies as well as large reasoning models.[‡]

## 1 Introduction

Large Language Model (LLM) scaling has recently undergone a paradigm shift toward increasing the amount of compute used during inference [1, 2], moving beyond traditional axes such as model size, training data, and pretraining compute [3, 4]. Scaling inference-time compute is particularly important for reasoning tasks, and is a key ingredient in OpenAI's o-series models [5], DeepSeek-R1 [6] among other frontier models [7, 8, 9, 10].

Despite the impressive performance of these systems, the central question of how to optimally allocate inference-time compute is not yet settled. The main challenge is that the space of strategies that use compute at test time is large and diverse: a wide variety of methods exist [2, 11, 12], each with its own empirical scaling law [1, 2]. Additionally, different methods can sometimes be combined, which further complicates any analysis.

In this paper, we seek fundamental and general principles that help clarify the landscape of inference-time compute. Since there is a large range of inference-time methods, in order to make progress we categorize methods into two classes [7]: (1) *parallel scaling* and (2) *sequential scaling*. We review these notions below.

---

[*]Equal contribution.

[†]Currently at Apple.

[‡]Code is available at https://github.com/seyedparsa/let-me-think.

39th Conference on Neural Information Processing Systems (NeurIPS 2025).

(1) **Parallel scaling** refers to generating multiple independent responses in parallel, and aggregating them in some way to output the final solution [13, 14, 15]. The most common aggregation technique is "best-of-$n$", where a reward function (e.g. another language model or a task-specific verifier [14]) selects the single highest-scoring response as the output. Another widely used aggregation method is majority voting, which determines the final response by choosing the most frequent one among all generated responses [13].

(2) **Sequential scaling** encompasses all techniques that do not fall under parallel scaling. The flagship method in this category is Chain of Thought (CoT) [16, 17, 18], in which an LLM first outputs a chain of reasoning tokens, before outputting its final answer. This may be achieved with one of several strategies to induce longer chains of reasoning in LLMs, such as adding a prompt instruction to "think step by step" [18], or forcing a longer chain of thought by replacing end-of-text tokens with "Wait" [7], or training with reinforcement learning objectives which can automatically induce longer chains of thought [6].

Consensus has yet to be reached on how to balance both types of scaling most effectively. On the one hand, sequential scaling via long chains of thought has demonstrated particular promise for tackling challenging problems, such as mathematics and coding benchmarks [19, 20, 21, 22, 6, 7]. On the other hand, the computational cost of inference grows quadratically in the context window for transformer-based architectures [23], making sequential scaling more expensive per-token than parallel scaling. This motivates the main question addressed in this work:

*Can we quantify the trade-off between sequential and parallel scaling for reasoning problems?*

## 1.1 Our contributions

Our main contribution is to introduce a reasoning task in which sequential scaling can be exponentially more powerful than parallel scaling. Namely, a small decrease in sequential scale necessitates a large increase in parallel scale to achieve the same level of accuracy. This tradeoff is illustrated in Figure 1 for transformer models evaluated on this task.

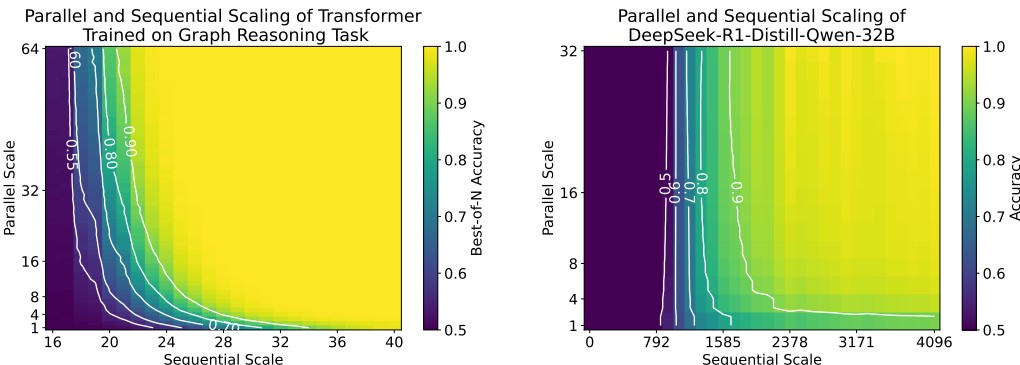

Figure 1: Model accuracy with different combinations of parallel and sequential scaling on a graph reasoning task. The sequential scale is the length budget for the chain of thought, and the parallel scale is the number of independent chains of thought generated. Left: The performance of a small transformer model trained for this task, where aggregation is with best-of-$n$. Right: The performance of the frontier DeepSeek-R1-Distill-Qwen-32B reasoning model [6], where aggregation is with majority vote. In both cases, there is a regime in which large increases to parallel scaling are required to compensate for a small decrease in sequential scaling; more details in Section 4.

**Reasoning task** Our reasoning task is a variant of the basic graph connectivity task. Solving it requires determining whether pairs of vertices are connected by stepping through several edges, and thus it serves as a proxy task modeling multi-step reasoning with CoT on more naturalistic data. Details are in Section 2.

Our task is motivated by a growing theoretical literature on the limitations and capabilities of transformers on graph reasoning tasks [24, 25, 26], which have found that graph connectivity is

challenging for bounded-depth transformers [27, 28, 29, 30, 31]. The reason for this is that graph computation appears to be a sequential problem, yet the transformers' sequential computation is bounded by their depth, as proved in expressivity results [32, 33, 34].

**Theoretical separations between sequential scaling and parallel scaling**    We consider bounded-depth, bounded-precision transformers on the connectivity problem, and we present two theoretical results that crystallize the intuition that graph connectivity requires sequential scaling which cannot be cost-effectively compensated by parallel scaling.

First, we prove that (a) sequentially scaling with one polynomial-length CoT can solve the connectivity problem, but in contrast (b) parallel scaling by aggregating over polynomially-many $O(1)$-length Chains of Thought cannot succeed. The proof of this theorem leverages recent results on the expressivity of transformers [32, 34], and requires making complexity-theoretic assumptions; see Section 3.1.

Second, in order to obtain a more fine-grained picture of the landscape and understand the performance of chains of thought with greater than constant length, we abstract transformer computation on graph reasoning tasks with a "Vertex Query Model" of computation. This model of computation is inspired by known limitations of transformers for multi-hop reasoning [31]. The Vertex Query Model has the benefit that it is tractable to analyze. Thus, we use it to guide the construction of challenging distributions of "two-path" graphs and "bridge" graphs, for which we give evidence that there is an exponential gap between the performance of sequential and parallel scaling; see Section 3.2.

**Experimental validation and exploration**    In Section 4, we empirically validate the challenging distributions of "two-path" and "bridge" graphs motivated by our Vertex Query Model. We use these tasks to test transformer-based language models trained to solve graph connectivity, and find that there is a significant advantage to scaling sequential computation over scaling parallel computation. We then extend this empirical investigation to leading open-source reasoning models, evaluating their performance on the graph connectivity task as well as on the more complex AIME2024 [35] dataset. The results reveal a consistent trend favoring sequential scaling over parallel scaling.

In Section 5, we explore training transformers on the graph connectivity problem with reinforcement learning (RL). We observe the emergent behavior that RL training gradually increases the length of the CoT. This behavior mirrors the growth in CoT length that occurs when training DeepSeek-R1 [6] with RL, and supports that the graph connectivity problems studied in this work are a rich enough task to capture many interesting behaviors observed in practice.

## 2   Graph reasoning tasks

Motivated by recent work on the expressivity and limitations of constant-depth transformers [28, 32, 31, 24, 29, 30], we test models on a graph connectivity task that serves as a basic testbed for reasoning. The most canonical connectivity task that one could consider is $(s, t)$-connectivity, defined below.

**Definition 1** ($(s, t)$-connectivity problem)**.** *The $(s, t)$-connectivity problem is: given a graph $G$ and vertices $s, t$ in this graph, return whether $s$ and $t$ are connected.*

One drawback of this task is that it is asymmetric – in the case that $s$ and $t$ are connected, there is a path certifying that they are connected. On the other hand, when $s$ and $t$ are in distinct components, there is no such path certificate. In order to ease our theoretical analysis and the experiments, we instead consider a more symmetrical problem that we call $(s, t_1, t_2)$-connectivity, which captures the essence of the difficulty in graph connectivity. We define this problem below.

**Definition 2** ($(s, t_1, t_2)$-connectivity problem)**.** *The $(s, t_1, t_2)$-connectivity problem is: given a graph $G$ and vertices $s, t_1$, and $t_2$ in this graph, return whether $s$ is connected to $t_1$ or $s$ is connected to $t_2$, given the promise that exactly one of these two alternatives is true.*

The benefit of this formulation of the problem is that in all cases, there is a path certifying the correct solution. For example, in the case that $s$ is connected to $t_1$, then the model can easily verify this in its chain of thought by finding a short path connecting $s$ to $t_1$.

Our theoretical results and our experiments are for $(s, t_1, t_2)$-connectivity in the setting where $G$ consists of two identical, disconnected components, one of the components contains $s$ and $t_i$, and the

other component contains $t_{3-i}$. The task is inputted as a list of edges, followed by the IDs of $s, t_1$, and $t_2$. See Figure 2 for an example of the input format.

# 3 Theoretical evidence for benefits of sequential over parallel scaling

We provide two main pieces of theoretical evidence for the benefits of sequential scaling over parallel scaling on these graph reasoning problems. We first prove a result based on expressivity limitations of bounded-depth transformers. Next, we obtain a more fine-grained picture based on an abstraction for CoT on graph reasoning problems that we call the vertex query model of computation.

## 3.1 Separation based on transformer expressivity limitations

We consider the $(s, t_1, t_2)$-connectivity problem on undirected graphs, as defined in Definition 1, where the size of the problem is given by the number of nodes $n$ in the graph. We study the most extreme case of parallel versus sequential scaling: many chains of constant length, compared to one long chain of polynomial length.

We leverage recent results on the expressive power of transformers with chain-of-thought to prove the following theorem. It requires making the complexity theory assumption that $\mathsf{TC}^0 \not\supseteq \mathsf{L}$, which is explained in Appendix A.

**Theorem 1** (Informal statement of Theorem 4). *Assume the complexity-theoretic statement that* $\mathsf{TC}^0 \not\supseteq \mathsf{L}$. *Then the following is true for bounded-depth, limited-precision transformers.*

- ***Sequential scaling succeeds****: There is a constant $c > 0$ such that a transformer with a CoT of length $\leq n^c$ solves any $(s, t_1, t_2)$-connectivity problem.*

- ***Parallel scaling fails****: For any constants $C_1, C_2 > 0$, and any transformer architecture, majority vote over $\leq n^{C_1}$ independently-sampled CoTs of length $\leq C_2$ has accuracy $\leq \frac{1}{2} + o(1)$ for $(s, t_1, t_2)$-connectivity problems.*

The above result may be rephrased as follows: parallel scaling requires at least a super-polynomial number of chains of thought of length $O(1)$ in order to simulate the computation achievable by sequentially scaling one chain of thought with polynomial length.

**Proof ingredients**  In Appendix A we provide the formal statement of the theorem and the full proof of the theorem. For the positive result, the main ingredient is from [32], which implies that transformers with polynomial length CoT can implement any polynomial-time algorithm, and therefore can implement breadth-first search which solves the connectivity problem. For the negative result, the expressivity bounds of [34, 33] imply that transformers with $O(1)$-length chain-of-thought fall into the class of circuits $\mathsf{TC}^0$. Our main insight is that aggregating multiple independently-sampled CoTs is also a $\mathsf{TC}^0$ circuit, and therefore is unable to solve $(s, t)$-connectivity under the complexity-theoretic assumption. Finally, we reduce from the $(s, t)$-connectivity problem to the $(s, t_1, t_2)$-connectivity problem with a $\mathsf{TC}^0$ reduction.

## 3.2 Evidence for separation based on the vertex query model

While the result in Theorem 1 is based on expressivity limitations of transformers, it is crude in the sense that (1) it does not provide a polynomial versus exponential separation, and (2) the parallel scaling limitations apply only to CoT of length $O(1)$. We now complement Theorem 1 with a more fine-grained lens on the tradeoff between sequential and parallel scale. In order to achieve this fine-grained result, we make a simplifying abstraction on the dynamics of Chain of Thought called the *Vertex Query Model (VQM)*. This computational model is more amenable to analysis than studying the $\mathsf{TC}^0$ circuit class.

**Definition 3** (Vertex Query Model). *An algorithm for $(s, t_1, t_2)$-connectivity is implementable in the Vertex Query Model (VQM) if it takes as input $s_1, t_1, t_2$, and can only access the graph $G$ through "neighborhood queries" $N_G$, which given a vertex $v$ output the set $N_G(v) = \{u : \exists(v, u) \in E\}$.*

*We also define the Restricted Vertex Query Model (RVQM), where the algorithm can only initially query $s$, and subsequently can only query vertices in the sets returned by previous queries.*

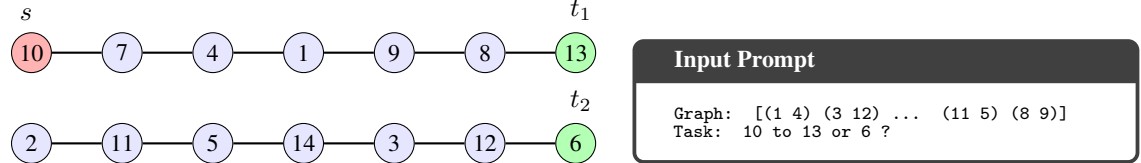

Figure 2: Left: an example "two-path" graph task from Theorem 2. Right: the task input is a list of edges in randomized order and with randomly permuted vertex IDs.

For the results in this section, we work under the simplifying abstraction that transformers with chain of thought are constrained to learning functions computable in the VQM with a cost at most proportional to the length of the chain of thought.

This simplifying abstraction is motivated by prior literature. First, constant-depth transformers are known to have limited range for multi-hop reasoning in graphs [31]. In chains of thought that output a sequence of nodes, it is reasonable to expect that the next node outputted by a transformer should lie only in a constant-depth neighborhood of the previous nodes or be a randomly-chosen node. These kinds of chains of thought correspond exactly to VQM algorithms. Second, the VQM is closely related to the previously proposed "globality barrier" for transformers learning to reason (Definition 3 of [28]). The "globality barrier" suggests that transformers with CoT can only efficiently learn functions such that each CoT step does a local computation – depending on a constant number of entries in the previous chain of thought. The VQM corresponds to such algorithms where the local computations allowed are neighborhood queries.

While the above arguments for VQM capturing the power of chain of thought are only heuristic, we now show that it is a useful abstraction because it motivates challenging families of graphs for the $(s, t_1, t_2)$-connectivity task. In Theorem 2 below, we show that algorithms in the VQM fail on problems where the graph is given by two disjoint paths, unless a large number of queries proportional to the length of the path is taken (corresponding under our assumption to a long chain of thought proportional to the length of the path).

**Theorem 2** (Minimum number of VQM queries needed for graph connectivity). *Consider the graph $G$ given by two disjoint paths of length $L \geq 3$ with randomly permuted vertex IDs. Suppose $s, t_1, t_2$ are distinct endpoints of these paths such that $s$ and $t_i$ are on the same path for exactly one $i \in \{1, 2\}$. Then*

- *$O(L)$ **queries are sufficient:** There is a VQM algorithm that executes $L - 1$ queries and solves the $(s, t_1, t_2)$-connectivity problem with probability 1.*

- *$\Omega(L)$ **queries are needed:** For any VQM algorithm that executes $q \leq (L - 2)/2$ queries, the probability of correctness of the algorithm on $(s, t_1, t_2)$-connectivity is exactly $1/2$.*

The proof is deferred to Appendix B. An example two-path graph is visualized in Figure 2. We experimentally validate in Figure 10 that, on frontier reasoning models, a minimal amount of sequential scale is needed to solve this problem, below which parallel scaling with majority vote is ineffective.

A drawback of the above theorem is that it proves limitations when the number of queries is smaller than the length of the shortest path between $s$ and $\{t_1, t_2\}$. In those situations, it may be impossible for algorithms in the VQM model to certify which $s$ and $t_i$ are connected. This raises the question: are there graphs where sequential scale is still beneficial even with more queries than the shortest path length? We provide one such example below, with the "bridge graph" construction. An example of this graph structure is illustrated in Figure 3.

**Definition 4** (Bridge Graph). *A bridge graph is an undirected graph parametrized by the non-negative integers* depth, short, long, *and* deadend. *It is constructed as follows:*

*Let $v_1 = s$ be the start node. Then, for each $i \in 1, \ldots$ depth $- 1$,*

1. *Let $v_{i+1}$ be the next "intersection"*
2. *Add two paths between $v_i$ and $v_{i+1}$, one of length* short *and the other* long
3. *Add a path from $v_i$ of length* deadend *(do not connect this to $v_{i+1}$)*

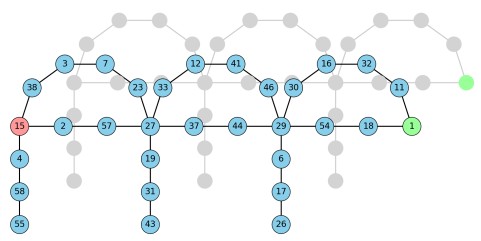

**Input Prompt**

```
Graph:  [(29 54) (15 2) ... (47 9) (32 16)]
Task:   15 to 8 or 1 ?
```

**Chain-of-thought**

```
DFS: [15 4 58 55 2 57 27 33 12 41 46 29 54 18 1]
Decision:  [1]
```

Figure 3: Left: Example "Bridge" graph task from Definition 4. Top right: the task input is a list of edges in randomized order and with randomly permuted vertex IDs. Bottom right: examples of chain of thought strategies used to train the model in our experiments in Section 4.

We now show that in the Restricted VQM, there is still a gap between sequential and parallel scale, even in a regime where more queries are made than the length of the shortest path in the bridge graph. Namely, with even a number of queries a constant fraction larger than the shortest path in this graph, any RVQM algorithm will be exponentially unlikely to succeed.

**Theorem 3.** *Consider an algorithm in the Restricted Vertex Query Model solving* $(s, t_1, t_2)$-*connectivity on the union of two identical copies of the Bridge$(d, l, 2l, 0)$ graph, where $s$ is the starting node on one side of the graph, and $t_1$ and $t_2$ correspond to the copies in the two connected components of the end node of the main path on the other side. For any $\delta \in (0, 1)$,*

1. ***Sequential scaling succeeds****: There exists an algorithm which makes $(1 + \delta)2ld$ queries and succeeds with probability at least $1 - \exp\left(-\frac{1}{2}d\delta^2\right)$*

2. ***Parallel scaling fails****: Any algorithm which makes no more than $(1-\delta)\frac{3}{2}ld$ queries succeeds with probability at most $\frac{1}{2} + \exp\left(-\frac{1}{2}\delta^2\frac{3}{2}d\right)$. Thus, parallel scaling with majority vote needs $\exp(\Omega(d))$ independent runs to succeed with probability $\geq 2/3$.*

We emphasize that the shortest path between the $s$ and $t_i$ is of length $ld$, and the theorem proves that any algorithm in the RVQM has exponentially poor advantage over random guessing even for a number of queries a constant fraction larger than this shortest path. The intuition is that each time the model hits an intersection (that is, a vertex with degree greater than two), it has to guess where to go next, and only has a constant probability per intersection of choosing the shortest path. The proof can be found in Appendix B.

## 4 Empirical evidence for benefits of sequential over parallel scaling

In this section, we experimentally study whether it is more efficient to parallelize multiple short CoTs or to scale one CoT sequentially. We validate the theoretical evidence put forward in Section 3 by (a) training transformer language models from scratch, and (b) evaluating leading open-source LLMs on the $(s, t_1, t_2)$-connectivity task with the "bridge graphs" of Theorem 3. Further experiments on the "two-path" graphs of Theorem 2 are available in Appendix C.

### 4.1 Chain of Thought strategies

In the transformers that we train from scratch, we seek to most efficiently use the chain-of-thought budget in order to best exhibit the full power of sequential scaling. In order to achieve this, we first train models on datasets generated by different CoT strategies, and then focus on the CoT strategy that has the best performance.

The CoT strategies that we consider provide a "proof" in the form of an exploration of the graph from the source to a sink, such as a path from the $s$ to either $t_1$ or $t_2$. This enables us to implement best-of-$n$ parallel scaling with a verifier for the proof. Even subject to the restriction of providing a

proof, the CoT lengths can still be short enough that we find that models trained on them get only barely higher than trivial accuracy.

The strategy with the shortest CoT that we consider is **Shortest-Path**, where the training data consists of shortest paths from the source node to the target node. Two other CoT strategies are derived from the trace of a depth-first-search (DFS) starting from the source node and ending at the target node. **Path** CoT is the path from the source node to the target node in the DFS tree, and **DFS** CoT is the list of DFS tree nodes ordered by when they are first visited in the DFS trace. The CoT and the final decision are appended to an input prompt to form a training example of the CoT strategy dataset (See Figure 3).

## 4.2 Evaluation metrics

Given a task as an input prompt, a model trained with a CoT strategy autoregressively generates a sequence of tokens either by greedy decoding or by sampling with a temperature. We extract the CoT and the decision from the output and evaluate each separately using the following criteria:

1. Decision Criterion: This checks if the decision is equal to the reachable target node.
2. Evidence Criterion: This verifies that the CoT starts with the source node, ends with one of the target nodes, and for every node in the CoT other than the source node, at least one of its adjacent nodes appears earlier in the CoT.

Building on these, we consider the following aggregation methods for evaluating parallel scaling:

1. Majority Decision: This takes the majority over the decisions of the sampled outputs.
2. Best-of-$n$: This checks if any of the sampled CoTs meets the evidence criterion, and if finds one, outputs its corresponding decision. Otherwise, it chooses one of the two target nodes at random as its decision.

We also define decision accuracy and evidence accuracy based on the decision and evidence criteria respectively, and evaluate models using them, over a set of test tasks from the same distribution as the training tasks. For parallel scaling evaluation, we compute decision accuracy for majority decision and best-of-$n$ methods.

## 4.3 Experiment setup

In our experiments, we let `short=3`, `long=5`, `deadend=3`, and refer to `Bridge(d, 3, 5, 3)` as `Bridge(d)`. To construct a task of depth $d$, we generate two randomly labeled `Bridge(d)` graphs, select the first starting node of one of the graphs as the source node $s$, and the last ending nodes of the two graphs as the target nodes $t_1$ and $t_2$. Finally, to transform the task into a sequence, we list the edges of the graphs in a random order, along with the labels of the source and target nodes as illustrated in Figure 3. For each CoT strategy and `Bridge(d)` task with depth from 1 to 5, we train a Mistral causal language model [36, 37] with 4 hidden layers, 4 attention heads, and intermediate size 128 with a context length of 400 for 200 epochs. In the experiments of each task, we use the same number of training tokens for all CoT strategies, equal to the number of training tokens in 500,000 samples from the Shortest-Path CoT strategy.

## 4.4 Results

We have found that models trained on DFS traces exploit a long CoT budget to outperform models trained on short CoTs (those generated by Shortest-Path and Path). The models trained on short CoTs tie with the DFS trained ones on very short budgets (at relatively low accuracy), but fall behind and plateau when given a higher token budget, as if they don't succeed early on, they don't know how to continue (which makes sense, since they have left their training distribution). The DFS model achieves perfect evidence and decision accuracy on the tasks, while the Path and Shortest-Path models struggle with the tasks when increasing the graph's depth; achieving 11.16% and 0.0% evidence accuracies respectively on the `Bridge(5)` task (See Figure 4).

**How models trained on short CoTs behave.** What scenarios at inference time lead to the failure of models trained on short CoTs? Looking more closely at the evidence accuracy of the models and

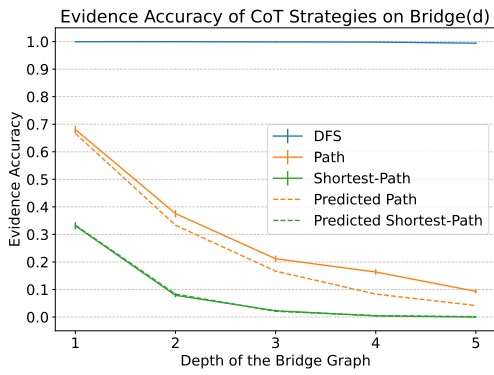 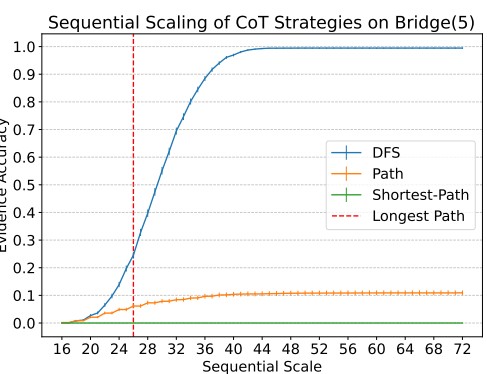

Figure 4: (left) Evidence accuracy of CoT strategies on `Bridge` tasks of various depths compared to the probabilities of a DFS trace becoming the shortest path and a path respectively. (right) Evidence accuracy of CoT strategies with different sequential CoT budgets on `Bridge(5)` task. Models outputs are sampled with greedy decoding. Error bars represent 95% binomial confidence intervals.

their behavior in response to an input, we find that although the Shortest-Path model is trained to take the short path from the starting node of each component to its end node, it cannot distinguish between the unexplored paths attached to the current node and gets into out-of-distribution scenarios by following the wrong paths and fails to recover from them. Therefore, given its limited CoT budget, its accuracy matches with the exponentially small probability $P(\text{DFS} \in D(\text{Shortest-Path})) = \frac{1}{3 \times 4^{d-1}}$ of a DFS trace that randomly chooses an unvisited adjacent node at each step, traversing the shortest path (See Figure 4). This supports the assumption that the model's limited expressivity limits its look-ahead ability, which motivated the Vertex Query Model (VQM) of Section 3.2. The Path model's accuracy follows a similar trend, but it is slightly higher than the in-distribution exploration probability $P(\text{DFS} \in D(\text{Path})) = \frac{2^d}{3 \times 4^{d-1}}$. The Path model's more flexible CoT budget and its ability to follow edges allows it to recover from some of the out-of-distribution scenarios it gets into by backtracking from the deadend or backward paths.

**Parallel scaling of models trained on short CoTs.** Since Figure 4 shows that sequential scaling of one chain of thought increases the accuracy significantly, we now ask: can we aggregate multiple short CoTs (either with best-of-$n$ or with majority voting) to achieve the same accuracy as one longer CoT? If so, how many short CoTs must we aggregate? We experiment by generating many short CoTs with temperature 1.0 and measuring accuracy for both majority and best-of-$n$ aggregation methods. For one run, decision accuracy is usually higher than evidence accuracy, because the model can both rely on its CoT to make a decision and if the CoT is not a valid proof it can randomly guess between the two target nodes. This also makes decision accuracy less robust when the model's evidence accuracy is low (See Figure 13 for evidence of both behaviors in short CoT models). However, when parallel scaling, the best-of-$n$ method that uses CoTs scales better than taking majority over the decisions (See Figure 7). Hence, we report the best-of-$n$ accuracy for the experiments with parallel scaling. We find that the best-of-$n$ accuracies of the Shortest-Path and Path models follow the probability that at least one of their $n$ independent sampled CoTs succeed, with the success probability of each corresponding to the exponentially small probability of traversing the shortest path and a path respectively (See Figure 7). Therefore, we need to sample an exponential number of CoTs from these models to achieve an accuracy on par with a single CoT of our models trained on long CoTs.

**Sequential scaling of CoT models.** Inspired by the observation that our short CoT models behave like the search models but with limited sequential CoT budget, we examine the evidence accuracy of each model with different sequential scales of CoT budget. We budget-force [7] the models by considering their CoTs of various maximum lengths, and find that at every sequential CoT budget, the DFS model achieves the highest evidence accuracy (See Figure 4). Then, to examine the best accuracy we could get with parallel scaling our models within a fixed sequential budget, we parallel scale the DFS model of different sequential scales. We find that sequential scaling up to a certain threshold is more effective than exponential parallel scaling (See Figure 1). Even from that threshold,

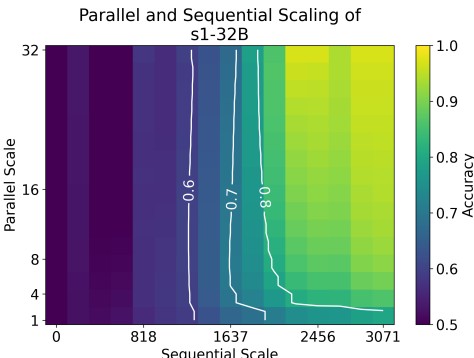 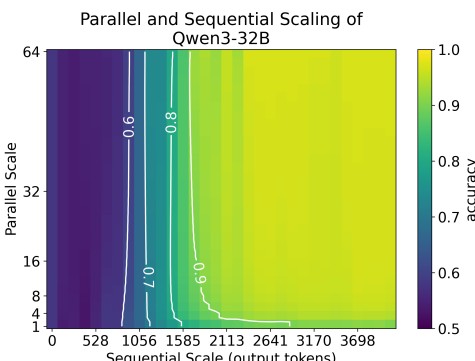

Figure 5: A comparison of parallel and sequential scaling for s1-32B [7] and qwen3-32B [9]. Note that the trend from Figure 1 is repeated. Sequential scaling is basically essential to get higher accuracy, and parallel scaling only becomes useful once sequential scaling has allowed for non-trivial performance.

scaling sequentially further is more efficient in terms of the number of tokens generated than parallel scaling (See Figure 8). In other words, for a fixed total token budget, sequential scaling always beats parallel scaling.

## 4.5 Experiments with large language models

We measured the performance of various LLMs on the graph connectivity task, as well as on the AIME2024 [35] benchmark.

Specifically, we consider the graph connectivity task on a bridge graph with `short=3, long=9, deadend=0, depth=2`. Mirroring our other experimental and theoretical results, LLMs only get trivial performance without a CoT, but when allowed a long CoT, they can achieve very high performance. We obtained similar trends with each of the three LLMs we tested, Qwen3-32B[9], DeepSeek R1 Distill Qwen-32B[6], and S1-32B[7] (See Figure 1 and Figure 5). Note that it takes roughly a thousand tokens of sequential scaling to get non-trivial accuracy, many of these tokens are used up by the LLM describing what its general approach to the problem will be, before actually executing a strategy.

We also conducted experiments with the s1-32B model [7] on the AIME2024 [35] dataset. The results show that sequential scaling cannot be efficiently replaced by parallel scaling for this mathematical task, supporting the generalizability of our findings to real-world scenarios (See Figure 11). While quantifying the exact trade-off between them for complex mathematical problems such as this is beyond the scope of our fundamental study, we observe that the results confirm our conclusion that sequential scaling is necessary.

## 5 Emergent sequential scaling with reinforcement learning

We observed that our models trained on short CoTs cannot look ahead to distinguish the correct and wrong paths and get into out-of-distribution scenarios. However, there are cases where the Path model recovers from the out-of-distribution scenario and succeeds in generating a verified CoT followed by the correct decision. This results in verified CoTs that are longer than any CoT in the training data (See Figure 4), with behaviors such as backtracking that are not present in the training data. How does reinforcing the model on its own verified CoTs, including these out-of-distribution CoTs, affect the model's performance and reasoning behavior? In this section, we explore this using Self-Taught Reasoner (STaR) [38], an expert iteration RL method, to fine-tune the model on its verified CoTs.

**Experiment setup** We perform a few iterations of STaR, where at each iteration, we sample responses to $500,000$ examples of the `Bridge(d)` task from the Path model with temperature $1.0$. Then we fine-tune the model for 20 more epochs on its own verified CoTs.

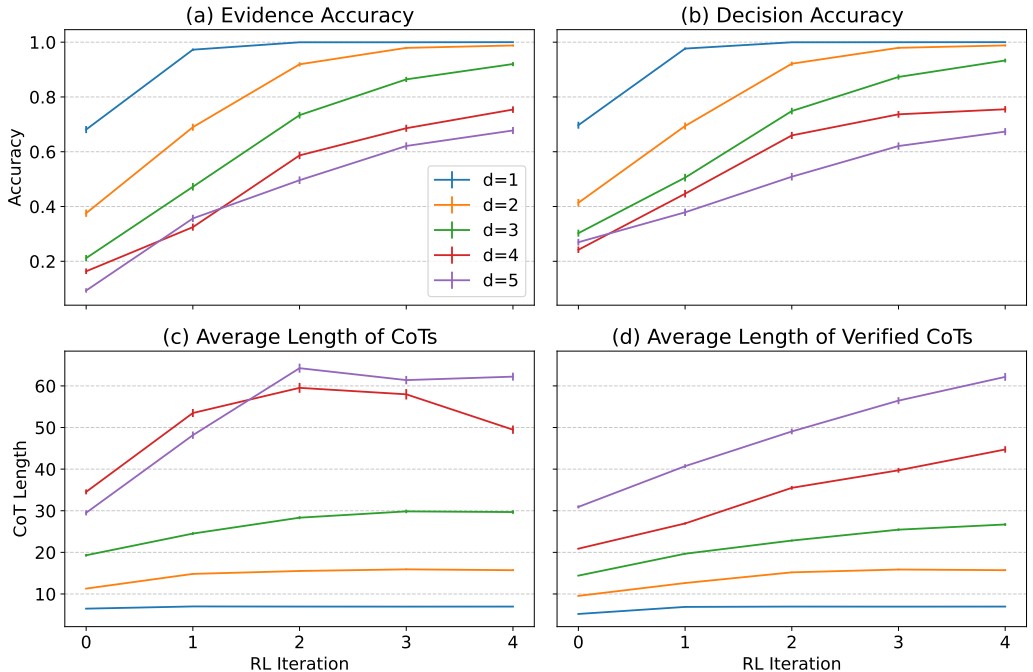

Figure 6: Path model's (a) Evidence accuracy, (b) Decision accuracy, and average length of (c) CoTs that follow the format, and (d) CoTs that are verified on `Bridge` tasks of various depths, before and after RL iterations. Error bars represent 95% binomial confidence intervals for accuracies, and 95% normal confidence intervals for CoT lengths.

**Results** We find that the model's accuracy on each task improves dramatically after a few iterations; evidence accuracy of the model pre-trained on `Bridge(3)` task jumping from 21.16% to 92.02% after 4 iterations. At the same time, the average length of the model's valid CoTs and verified CoTs increases and the model learns to exploit increasingly longer CoTs after each RL iteration (See Figure 6). Moreover, we find that the model's accuracy improves at every sequential CoT budget (See Figure 9). This gives insight into the observed phenomena of long CoT emergence during RL on reasoning tasks [6, 39, 40]. RL can adapt the model's training to its expressivity for the task, by reinforcing its own computations that result in solving the task. In our case, the model is not expressive enough to solve the task by following the CoT strategy it was trained on. However, after training on its successful generations during RL, its CoT scales sequentially to follow a longer but more simple strategy it is expressive enough to adopt.

## 6 Discussion

Our results on graph connectivity demonstrate that there are settings in which sequential scaling is vastly more cost-effective than parallel scaling. However, our results are limited only to the setting that we study, and the optimal recipe for test-time compute may be problem-dependent, lying in a mixture of combining both parallel scaling and sequential scaling. Indeed, our experiments indicate that once the sequential scale becomes large enough, parallel scaling becomes a more cost-effective axis to scale due to diminishing returns to sequential scaling. Understanding the general principles that determine the optimal mix of parallel and sequential scaling for a given dataset is an interesting direction for future study. Additional discussion of limitations beyond what is presented in this section, as well as discussion of further related work, can be found in Appendix D and Appendix E.

## Acknowledgments

This work was initiated while EB, EE, EM, and PM were visiting the Simons Institute for the Theory of Computing. EB was supported by the Simons Institute as a Research Fellow at the Special Year on Large Language Models and Transformers, and also by NSF grant CCF-2106377. EE acknowledges a gift from AWS AI to Penn Engineering's ASSET Center for Trustworthy AI. PM acknowledges support from the National Science Foundation (NSF), the Simons Foundation for the Collaboration on the Theoretical Foundations of Deep Learning, and the Office of Naval Research through awards DMS-2031883, #814639, and ONR-N000142412631. SJ acknowledges funding support by the Center of Mathematical Sciences and Applications. EM was supported by the Kempner Institute for the Study of Natural and Artificial Intelligence, which was made possible in part by a gift from the Chan Zuckerberg Initiative Foundation. This work used the Delta system at the National Center for Supercomputing Applications through allocation TG-CIS220009 from the Advanced Cyberinfrastructure Coordination Ecosystem: Services & Support (ACCESS) program, which is supported by National Science Foundation grants #2138259, #2138286, #2138307, #2137603, and #2138296 [41].

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

# A  Separation between exponential and sequential scaling based on expressivity

We prove Theorem 4, which is the formal version of the Theorem 1 stated in the main text.

## A.1  Preliminaries: expressivity of transformers

Before stating the theorem formally and proving it, let us first review known technical bounds on the expressivity of limited-precision, bounded-depth transformers [42, 43, 34]. To present these, we must first define the computational model of threshold circuits.

**Definition 5** ($\mathsf{TC}^0$ computational model). *A $\mathsf{TC}^0$ circuit is a boolean circuit with AND, OR, NOT, and MAJORITY gates of potentially unbounded fan-in. A $\mathsf{TC}^0$ circuit family is a collection of circuits indexed by the input size $n$, such that for each input size the circuit has polynomial width and bounded depth.*

It has recently been shown that constant-depth transformers can be well-approximated by the class of threshold circuits of constant depth.

**Proposition 1** (Transformers are in $\mathsf{TC}^0$; implied by Theorem 14 of [34]). *For any bounded-depth softmax-attention transformer $T : \Sigma^* \to \mathbb{R}^{|\Sigma|}$ and any polynomial $p(n)$, there is a function $\hat{T} : \Sigma^* \to \mathbb{R}^{|\Sigma|}$ in $\mathsf{TC}^0$ that approximates $T$ to $2^{-p(n)}$ additive error on inputs of length $n$.*[1]

This implies limitations on the expressive power of transformers, under standard computational complexity assumptions. In particular, it is a common conjecture that $\mathsf{TC}^0$ circuits are unable to determine $s$-$t$ connectivity in undirected graphs [44, 45], and this conjecture is normally stated as $\mathsf{L} \not\subseteq \mathsf{TC}^0$.[2], because $\mathsf{L}$ is a complete problem undirected graph connectivity [46, 47]. Therefore, Proposition 1 provides evidence that bounded-depth and poly-size transformers (without chain of thought) are not able to directly determine whether two nodes are connected in an inputted graph.

## A.2  Our result

Proposition 1 has not been shown to imply a tradeoff between parallel and sequential scaling in transformers, which is the new contribution in Theorem 4 proved in this section.

Given a function $T : \Sigma^* \to \mathbb{R}^{|\Sigma|}$ operating on a polynomial-size alphabet of tokens $\Sigma$, and an input prompt $x \in \Sigma^k$, we inductively define the autoregressive distribution

$$D_{T,n}(x)$$

formed by sampling $n$ tokens autoregressively from the transformer. $D_{T,0}$ is the empty string with probability 1. For any $n \geq 1$, the distribution $D_{T,n}$ is the distribution of $[z_1, \ldots, z_n]$ where $[z_1, \ldots, z_{n-1}] \sim D_{T,n-1}$, and $z_n \sim \mathrm{softmax}(T([x; z_1, \ldots, z_{n-1}]))$.

We first prove that the distribution of outputs from a transformer is close in total variation to one generated by iteratively applying a $\mathsf{TC}^0$ circuit.

**Lemma 1** (Approximating the autoregressive distribution of a transformer). *Given a transformer $T : \Sigma^* \to \mathbb{R}^{|\Sigma|}$ and polynomials $p_1(n), p_2(n)$, there is a function $\hat{T}$ in $\mathsf{TC}^0$ such that for all $x \in \Sigma^n$*

$$d_{TV}(D_{T,m}(x); D_{\hat{T},m}) \leq 2^{-p_1(n)} ,$$

*for any $m \leq p_2(n)$, where $d_{TV}$ denotes the total variation distance between distributions.*

*Proof.* Let $p(n)$ be a polynomial that we will fix later. Let $\hat{T}$ be a $\mathsf{TC}^0$ circuit family such that $\hat{T}$ approximates $T$ up to $2^{-p(n)}$ additive error on inputs of length $n$, as guaranteed by Proposition 1. For

---

[1]The $\mathsf{TC}^0$ circuit outputs in $\mathbb{R}^{|\Sigma|}$ is returned up to some number of bits of precision.

[2]For directed graphs, which we will not use here, the relevant conjecture is $\mathsf{NL} \not\subseteq \mathsf{TC}^0$

$m = 0$, we have $d_{TV}(D_{T,0}(x), D_{\hat{T},0}(x)) = 0$ by definition. For any string $s$ of length $\geq n$, we have

$$d_{TV}(D_{T,1}(s), D_{\hat{T},1}(s)) = \frac{1}{2} \sum_{i \in \Sigma} |\frac{\exp(T(s)_i)}{\sum_{j \in \Sigma} \exp(T(s)_j)} - \frac{\exp(\hat{T}(s)_i)}{\sum_{j \in \Sigma} \exp(\hat{T}(s)_j)}|$$

$$\leq |\exp((|\Sigma| + 1)2^{-p(n)}) - \exp(-(|\Sigma| + 1)2^{-p(n)})|$$

$$\leq 5(|\Sigma| + 1)2^{-p(n)},$$

whenever $n$ is large enough and $2^{-p(n)}(|\Sigma| + 1) \leq 1$. So combining with the data-processing inequality, for any $m \geq 1$, we have

$$d_{TV}(D_{T,m}(x), D_{\hat{T},m}(x))$$

$$\leq d_{TV}(D_{T,m-1}(x), D_{\hat{T},m-1}(x)) + \mathbb{E}_{z \sim D_{T,m-1}(x)}[d_{TV}(D_{T,1}([x; z]), D_{\hat{T},1}([x; z]))]$$

$$\leq d_{TV}(D_{T,m-1}(x), D_{\hat{T},m-1}(x)) + 5(|\Sigma| + 1)2^{-p(n)}.$$

Applying this inductively on $m$ yields

$$d_{TV}(D_{T,m}(x), D_{\hat{T},m}(x)) \leq 5m(|\Sigma| + 1)2^{-p(n)}$$

$$\leq 5p_2(n)(|\Sigma| + 1)2^{-p(n)}.$$

Choosing $p(n)$ large enough so that the right-hand side is $\leq 2^{-p_1(n)}$ concludes the proof. $\qquad\square$

This allows us to consider the autoregressive distributions generated by $\mathsf{TC}^0$ circuits, which we will find easier to analyze than the autoregressive distributions generated by transformers.

We observe that, for constant-length chains of thought, the autoregressive distribution is also directly sampleable by a $\mathsf{TC}^0$ circuit with no chain of thought. This lemma was effectively claimed in Figure 1 of [32], but without a proof.

**Lemma 2** (Constant-length CoT simulated by randomized $\mathsf{TC}^0$). *Let $C$ be a constant number of steps, and let $\hat{T} : \Sigma^* \to \mathbb{R}^{|\Sigma|}$ be a function in $\mathsf{TC}^0$. Define the distribution of the last token $\hat{P}(x)$ to be the law of $z_C$ where $z \sim D_{\hat{T},C}(x)$.*

*Then for any polynomial $p_1(n)$, there is a polynomial $p_2(n)$ and a function $\tilde{T} : (\Sigma \cup \{0, 1\})^* \to \Sigma$ in $\mathsf{TC}^0$ such that for all $x \in \Sigma^n$ we have*

$$d_{TV}(\hat{P}(x); \tilde{P}(x)) \leq 2^{-p_1(n)}$$

*where $\tilde{P}(x)$ is the law of $\tilde{T}(x; r)$, where $r \sim \mathrm{Unif}[\{0, 1\}^{p_2(n)}]$ are random input bits.*

*In other words, one step of $\tilde{T}$ approximates $C$ autoregressive steps of $\hat{T}$.*

*Proof.* For any polynomial $p(n)$, there is a $\mathsf{TC}^0$ circuit that (given a polynomial number of random bits), samples from a step of the autoregressive distribution with $\hat{T}$ up to total variation error $2^{-p(n)}$. This is because first the circuit can compute $\hat{T}$, and then the softmax operation can be approximated by $\mathsf{TC}^0$ circuits, as proved in Theorem 14 of [34]. Concatenating this circuit $C$ times, we obtain a randomized $\mathsf{TC}^0$ circuit $\tilde{T}$ that satisfies the lemma, as long as we take $p(n) \geq p_1(n) \log_2(1/C)$. $\quad\square$

Now recall the folklore result that $\mathsf{TC}^0$ circuits can be derandomized.

**Lemma 3** (Derandomization of $\mathsf{TC}^0$; folklore). *Let $p(n)$ and $p'(n)$ be polynomials and $\tilde{T} : (\Sigma \cup \{0, 1\})^* \to \Sigma$ be a $\mathsf{TC}^0$ function.*

*Then, there is a $\mathsf{TC}^0$ function $\dot{T} : \Sigma^* \to \Sigma$ such that for any $n$, any $x \in \Sigma^n$ and $\sigma \in \Sigma$, we have*

$$\dot{T}(x) = \sigma, \ if \ \mathbb{P}_{r \sim \{0,1\}^{p(n)}}[\tilde{T}(x; r) = \sigma] \geq 1/2 + 1/p'(n).$$

*Proof.* Let $p_1(n)$ be a polynomial that we will fix later. Consider the circuit $T'$ that upon input $[x; r_1, \ldots, r_{p_1(n)}]$ where $x \in \Sigma^n$ and $r_i \in \{0, 1\}^{p(n)}$, takes a majority vote over

$\tilde{T}(x; r_1), \ldots, \tilde{T}(x; r_{p_1(n)})$. By a Chernoff bound, and a large enough polynomial $p_1(n)$, we have that for any $x \in \Sigma^n$ and $r \in \{0,1\}^{p(n)}$, we have

$$\mathbb{P}_{r_1,\ldots,r_{p_1(n)}}[T'(x; r_1, \ldots, r_{p_1(n)}) = \sigma] \geq 1 - |\Sigma|^{-n-1} \text{ if } \mathbb{P}_{r \sim \{0,1\}^{p(n)}}[\tilde{T}(x; r) = \sigma] \geq 1/2 + 1/p'(n).$$

By a union bound over all inputs $x \in |\Sigma|^n$, for any $n$ there is a random seed $[r_1^*, \ldots, r_{p_1(n)}^*]$ such that

$$T'(x; r_1^*, \ldots, r_{p_1(n)}^*) = \sigma, \text{ if } \mathbb{P}_{r \sim \{0,1\}^{p(n)}}[\tilde{T}(x; r) = \sigma] \geq 1/2 + 1/p'(n).$$

For any $x \in \Sigma^n$, let $\dot{T}(x) = T'(x; r_1^*, \ldots, r_{p_1(n)}^*)$, which is in $\mathsf{TC}^0$ since the seed can be hardcoded into the circuit and is of polynomial length. $\qquad\square$

The final ingredient is a $\mathsf{TC}^0$ reduction from $(s,t)$-connectivity to $(s,t_1,t_2)$-connectivity.

**Lemma 4.** *Suppose that the function $f(G, s, t_1, t_2)$ solving $(s, t_1, t_2)$-connectivity instances is in $\mathsf{TC}^0$. Then $\mathsf{TC}^0 \supseteq \mathsf{L}$.*

*Proof.* We will show that if $f$ is in $\mathsf{TC}^0$, then $(s,t)$-connectivity is also in $\mathsf{TC}^0$. The reduction is as follows. Create a $(u, v_1, v_2)$-connectivity problem $(H, u, v_1, v_2)$ by letting $H = G_1 \sqcup G_2$ be a disjoint union of two copies of $G$. Randomly choose $i \in \{1, 2\}$, and let $u$ be the copy of $s$ in $G_i$. Let $v_1$ be the copy of $t$ in $G_1$ and let $v_2$ be the copy of $t$ in $G_2$. Also, permute the labels and the order of the edges by some permutation $\sigma$ that we will choose randomly. Finally, compute $a = f(H, u, v_1, v_2)$ and return true if $a = v_i$ and false otherwise. There are two cases:

- If $s$ and $t$ are connected in $G$, then $(H, u, v_1, v_2)$ is a well-formed $(u, v_1, v_2)$-connectivity problem, so $f(H, u, v_1, v_2)$ will always output $v_i$, and so the final answer is "true".

- If $s$ and $t$ are not connected in $G$, then over the randomness of the label and edge permutations the probability that $f$ returns $v_i$ is exactly 1/2 (because the component in which $v_1$ resides and the component in which $v_2$ resides are indistinguishable).

Finally, repeat this procedure in parallel with $\mathrm{poly}(n)$ different random permutations, and return "true" if the answer for all repetitions is "true", and "false" otherwise. By a union bound, over the set of possible inputs, there is a deterministic choice of $\mathrm{poly}(n)$ permutations such that this procedure is correct on any size-$n$ input $(G, s, t)$. This overall procedure can thus be implemented in $\mathsf{TC}^0$ by hard-coding those permutations into the circuit for any $n$.

Thus, we have shown a $\mathsf{TC}^0$ circuit for $(s,t)$-connectivity. Recall that $(s,t)$-connectivity is complete for the class $\mathsf{L}$ under $\mathsf{TC}^0$ reductions (see e.g., [44, 45]), so $\mathsf{L} \subseteq \mathsf{TC}^0$, concluding the proof. $\qquad\square$

With these preliminaries, we arrive at Theorem 4, which is the formal statement of Theorem 1, which was in the main text. We assume that there are two output tokens yes, no $\in \Sigma$, and the transformer's final token in the chain of thought is its response – either yes or no.

**Theorem 4.** *We have the following results for $(s, t_1, t_2)$-connectivity problems of size $n$ and transformers.*

- ***Sequential scaling succeeds**: There is a constant $c > 0$ such that a log-precision transformer with a CoT of length $\leq n^c$ solves any $(s, t_1, t_2)$-connectivity problem.*

- ***Parallel scaling fails**: Assume that $\mathsf{L} \not\subseteq \mathsf{TC}^0$. Let $C_1, C_2 > 0$ be constants, and let $T : \Sigma^* \to \mathbb{R}^{|\Sigma|}$ be a polynomial-precision transformer. Let $m(n) := n^{C_2}$ be the number of chains of thought over which we take majority vote (breaking ties arbitrarily). Then there are infinitely-many $n$ such that there is a size-$n$ $(s, t_1, t_2)$-connectivity graph problem $(G, s, t_1, t_2)$ with answer $\mathrm{ans} \in \{\text{yes}, \text{no}\}$, such that*

$$\mathbb{P}_{z_1,\ldots,z_{m(n)} \sim D_{T,C_1}(G,s,t)}[\mathrm{Majority}(z_{1,C_1}, \ldots, z_{m(n),C_1}) = \mathrm{ans}] < 1/2 + 1/n.$$

  *I.e., majority vote over $m(n)$ parallel chains of thought with length $C_1$ is correct with probability at most $1/2 + o(1)$.*

*Proof.* For the positive result that sequential scaling succeeds, it is sufficient to use Corollary 2.1 of [32], which implies that log-precision transformers with $t(n)$-length chain of thought can simulate Turing machines that run in time $t(n)$. Since $(s, t_1, t_2)$-connectivity is solvable in polynomial time (e.g. with breadth-first search), the first part of the theorem follows.

For the negative result that parallel scaling fails, we use the lemmas that we have developed above. Suppose by contradiction that for large enough $n$, we have for all size-$n$ problems $(G, s, t_1, t_2, \text{ans})$ that

$$\mathbb{P}_{z_1,\dots,z_{m(n)} \sim D_{T,C_1}(G,s,t_1,t_2)}[\text{Majority}(z_{1,C_1}, \dots, z_{m(n),C_1}) = \text{ans}] \geq 1/2 + 1/n.$$

Then by Lemma 1 with precision $1/n$, and by triangle inequality, there is a $\mathsf{TC}^0$ function $\hat{T}$ such that for all large enough $n$ and all size-$n$ problems $(G, s, t_1, t_2, \text{ans})$, we have

$$\mathbb{P}_{z_1,\dots,z_{m(n)} \sim D_{\hat{T},C_1}(G,s,t_1,t_2)}[\text{Majority}(z_{1,C_1}, \dots, z_{m(n),C_1}) = \text{ans}] \geq 1/2 + 2/n.$$

By Lemma 2 again with precision $1/n$, and by a triangle inequality, there is a $\mathsf{TC}^0$ function $\tilde{T}$ that approximates the autoregressively-applied $\hat{T}$, in the sense that there is a polynomial $\tilde{p}$ such that for any size-$n$ problem $(G, s, t_1, t_2, \text{ans})$

$$\mathbb{P}_{r_1,\dots,r_{m(n)} \sim \{0,1\}^{\tilde{p}(n)}}[\text{Majority}(\tilde{T}(x; r_1), \dots, \tilde{T}(x; r_{m(n)})) = \text{ans}] \geq 1/2 + 3/n.$$

Since Majority is a gate, the circuit $\text{Majority}(\tilde{T}(x; r_1), \dots, e T(x; r_{m(n)}))$ is a $\mathsf{TC}^0$ function and so it can be derandomized by Lemma 3. Using this lemma, yields a $\mathsf{TC}^0$ function $\dot{T}$ such that for any size-$n$ problem $(G, s, t_1, t_2, \text{ans})$,

$$\dot{T}(G, s, t_1, t_2) = \text{ans}.$$

Using Lemma 4, this implies $\mathsf{L} \subseteq \mathsf{TC}^0$, which contradicts our assumption that $\mathsf{L} \not\subseteq \mathsf{TC}^0$.

$\square$

# B   Evidence from vertex query model for sequential vs. parallel scaling separation

## B.1   Separation in vertex query model, Proof of Theorem 2

In the VQM, we can prove the necessity of a minimum number of queries (corresponding to a minimum length for a chain of thought by our simplifying abstraction that the VQM models the capabilities of transformers with bounded chain-of-thought).

**Theorem 5** (Minimum number of VQM queries needed for graph connectivity; restatement of Theorem 2). *Consider the graph $G$ given by two disjoint paths of length $L \geq 3$ with randomly permuted vertex IDs. Suppose $s, t_1, t_2$ are distinct endpoints of these paths such that $s$ and $t_i$ are on the same path for exactly one $i \in \{1, 2\}$. Then*

- $\Omega(L)$ *queries needed: For any VQM algorithm that executes $q \leq (L-2)/2$ queries, the probability of correctness of the algorithm on $(s, t_1, t_2)$-connectivity is exactly $1/2$.*

- $O(L)$ *queries sufficient: There is a VQM algorithm that executes $L - 1$ queries and solves the $(s, t_1, t_2)$-connectivity problem with probability 1.*

*Proof.* For the positive result, consider the algorithm that queries $s$, then the neighbor of $s$, and so on, until it reaches the other end of the path. This takes at most $L - 1$ queries, and reaches either $t_1$ or $t_2$, at which point the algorithm has enough information to return the correct answer.

For the analysis of the negative result, let $u_1, \dots, u_L$ denote the ordered vertices of the first path and let $v_1, \dots, v_L$ denote the ordered vertices of the second path. Let the algorithm run and make $q \leq (L-2)/2$ queries. By the pigeonhole principle there must be an $i \in \{1, \dots, L-1\}$ such that the algorithm has not queried $u_i, v_i, u_{i+1}$ and $v_{i+1}$. Now note that if we additionally reveal the neighborhoods of $u_1, \dots, u_{i-1}, u_{i+2}, \dots, u_L$ and $v_1, \dots, v_{i-1}, v_{i+2}, \dots, v_L$ with vertex queries then the algorithm still has probability of success $1/2$, since it is equally likely given its information that $u_i$ is connected to $u_{i+1}$ as it is for $v_i$ to be connected to $v_{i+1}$. $\square$

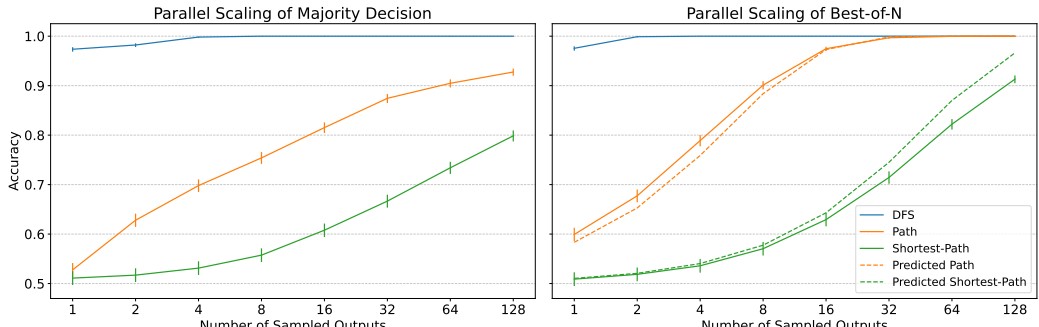

Figure 7: (left) Majority decision and (right) Best-of-$n$ accuracy for parallel scaling of models trained with CoT strategies for `Bridge(3)` task, compared to the accuracy predicted by each CoT independently meeting the evidence criteria with the probability of a DFS trace becoming the shortest path and a path respectively.

## B.2 Proof of Theorem 3

*Proof.* First we will show the lower bound.

Without loss of generality, assume that the model will explore from $s$, and stop when it reaches $t_1$ or $t_2$ (note that because the vertex labels are uniformly random, there is no other way of getting a higher than $50\%$ success rate than finding $t_1$ or $t_2$ when starting from $s$).

To get from $s$ to $t_b$, the algorithm must explore each intersection (those vertices with degree greater than two). To get from the current intersection to the next one, the algorithm has no way to distinguish between the long and short path until it explores at least $l$ vertices, and so there is at most a $1/2$ chance the model takes $l$ oracle calls to get to the next intersection, and at least a $1/2$ chance it takes $2l$ oracle calls (if it takes the long path for $l$ vertices, then any node it has discovered is still $l$ vertices away from the next intersection, so it must make at least $l$ more calls). Since there are $d$ intersections[3], a standard Chernoff bound for iid Bernoulli random variables shows that the probability of finding $t_b$ in at most $(1 - \delta)\frac{3}{2}ld$ oracle calls is at most $\exp\left(-\frac{1}{2}\delta^2 \frac{3}{2}d\right)$, and if we don't find $t_b$, then the best the algorithm can do is guess, and get a $1/2$ probability of being correct, yielding the desired result.

For the upper bound, we will consider this algorithm: each time we reach a new intersection (including the start), choose an unexplored neighbor, and explore down that path for $l$ vertices, and if the next intersection is not found, try one of the other unexplored paths from before.

At a new intersection, the algorithm has three unexplored paths:

1. The short path to the next intersection
2. The long path to the next intersection
3. The path to the previous intersection it didn't take

So, notice that the algorithm we defined has a $1/3$ chance of taking $l$ oracle calls to reach the next intersection, a $1/3$ chance of taking $2l$, and $1/3$ chance of taking $3l$. Using Hoeffdings inequality, the probability the algorithm takes more than $(1 + \delta)2ld$ oracle calls is at most $\exp(-2d\delta^2)$, so the algorithm succeeds with at least one minus this probability.

$\square$

## C Experimental details and further experiments

### C.1 Training

For each CoT strategy and task, we train a Mistral causal language model [36, 37] with 4 hidden layers, 4 attention heads, and intermediate size 128 with a context length of 400 for 200 epochs on NVIDIA A100 GPU with 40GB memory. We sweep through the learning rate values in {1e-4, 3e-4,

---

[3]Including $s$, for which the same logic applies when getting from $s$ to the next intersection.

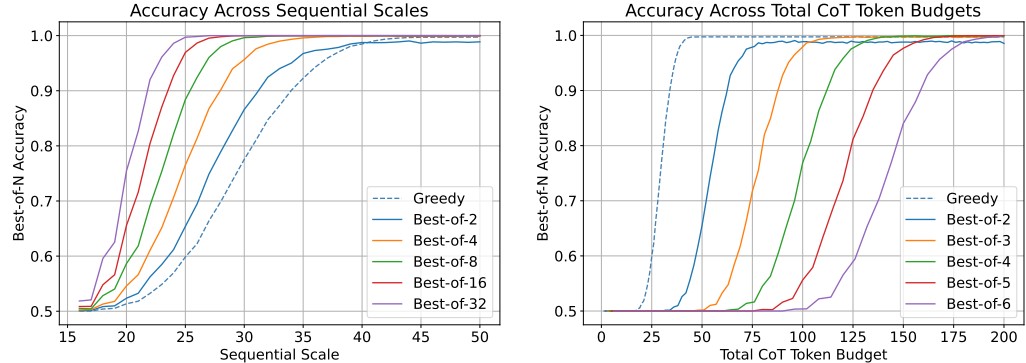

Figure 8: Best-of-N accuracy for parallel scaling of the model trained with DFS CoT strategy on `Bridge(5)` task (left) across sequential scales (maximum CoT length) and (right) total CoT token budget. Outputs are sampled with temperature 1.0 for parallel scaling.

1e-3, 3e-3} and train the model for 200 epochs with a batch size of 1000. We have also experimented with different weight decay values and learning rate schedules, but we found no significant difference in the results and used 0.05 weight decay and a cosine learning rate schedule, with a 0.1 warm-up ratio. We use the same hyperparameters for RL iterations, except that we fine-tune the model for 20 epochs at each iteration. Each pretraining experiment takes under 12 GPU hours, while fine-tuning for RL takes under 3 GPU hours. Additionally, debugging and hyperparameter tuning for each experiment took under 72 GPU hours.

## C.2 Sequential scaling of walk strategies

**Experiment setup** To study sequential scaling of CoTs in a controlled setting, we also ran experiments with a CoT strategy with tunable scale. A **Walk-L** CoT is generated by sampling a random walk that starts at the source node, conditioned on visiting the target node within at most $L$ steps. Hence, models trained with Walk-L strategies at different scales $L$ are exposed to successful traces of random walk on the same task, but with different number of steps the walk is allowed to take to reach the target. As $L$ increases, the CoTs become longer, less optimal, and more exploratory.

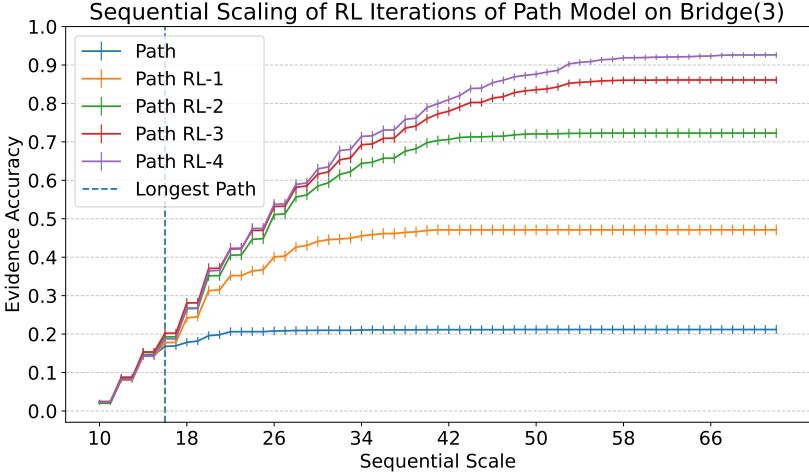

Figure 9: Evidence accuracy of Path model before and after RL iterations with different sequential CoT budgets on the `Bridge(3)` task. Error bars represent 95% binomial confidence intervals.

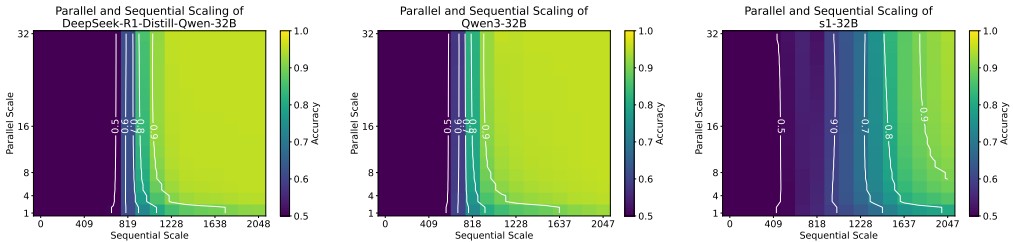

Figure 10: A comparison of parallel and sequential scaling for three LLMs tested on the $(s, t_1, t_2)$-connectivity problem for a graph that is the disjoint union of two paths. Note the similar trend to Figure 1.

**Results** After training models for the `Bridge(5)` task with Walk-L CoT strategies, we find that the accuracy of the models consistently increases with $L$, which shows that the models trained on more exploratory and longer walks perform better (See Figure 14).

## C.3 Experiment with a smaller transformer

**Experiment setup** We also ran experiments using smaller transformer models with 2 hidden layers, and a variant of DFS strategy called **DFS-BT**. In a CoT of DFS-BT strategy, we include the whole DFS trace, which is a walk in the DFS tree including the backtracking steps.

**Results** We find that small models trained on DFS-BT CoTs solve the task consistently, while small models trained on DFS CoTs fail to solve the `Bridge` tasks of larger depths (See Figure 13), which can be explained by the smaller model's more limited expressivity.

## C.4 LLM experimental details

For the AIME2024 experiment, we used H200 GPUs. Each run took approximately 1.5 hours, for a total of about 24 H200 GPU-hours. For the graph connectivity experiment, we used vllm and 2 A100 GPUs (80 GB of memory each) for inference. The experiments to make each plot took less than four hours each. Debugging and hyperparameter tuning took under 120 GPU hours. We constructed 32 random labelings of the bridge graph, and then, using prompts of the form Figure 12, create CoTs of 4096 tokens. Depending the model, we added the appropriate special tokens to make the input prompt from the user, and to make the model use thinking mode during the CoT. Each model recommended using temperature $0.6$ for thinking, which we did. We used a custom logit processor to make the model substitute the end thinking token and the eos token with the token for "wait", inspired by [7]. Then we truncate the CoT at intervals evenly spaced by tokens, and append the end of thinking token, and "Answer: Node [start node label] is in the same connected component as node " before using the model to find the logits for the next token. The model is considered correct if the logit for the correct node is higher than the logit for the incorrect node introduced in the initial prompt[4]. For parallel scaling, we generated up to $64$ distinct CoTs for each graph, and analytically calculated the probability that a random subsample would vote for the correct or incorrect solution (or tie). All of the results have a standard deviation of at most $0.08$.

## C.5 LLM additional experiments

In Figure 10 also tested the LLMs on a setting closer to the setting of Theorem 2, where the graph to be explored is two disjoint paths, and we once again confirm the theory, and see similar trends to those in Figure 1.

---

[4]With some tie breaking when the logits are within $1e{-}8$ of each other. We found that techniques weighting the confidence by the magnitude of the logits or their difference did not significantly change any results.

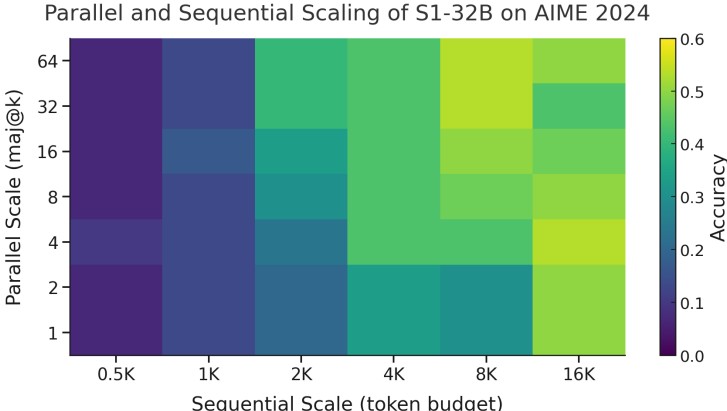

Figure 11: A comparison of parallel and sequential scaling for s1-32B [7] on AIME2024 [35]. For parallel scaling, answers are sampled with temperature 1.0 and aggregated by majority vote.

## D  Further discussion

While we make an effort to find the best models for graph connectivity with chain-of-thought (See Figure 4) in our experiments, we do not have a guarantee that these are indeed the best models that deploy chain of thought. In future work, this could be addressed by studying models learned with RL, with a penalty on the length of the chain of thought, to encourage more optimal use of the sequential scaling budget.

Additionally, the Vertex Query Model that we propose to abstract the power of chain-of-thought in Section 3.2 is motivated by the globality barrier studied in [28], and is empirically validated, but it does not have direct theoretical backing. An interesting future direction is to prove that bounded-depth transformers on graph connectivity tasks are indeed effectively restricted by this model.

## E  Further related work

**Expressivity of transformers with CoT**  The representational power of transformers has been studied in several works [31, 24, 33, 48, 49]. Recent work also highlights the expressivity and sample efficiency gains of reasoning with chain-of-thoughts [50, 51, 52, 53, 32, 54, 55]. In particular, many studies use graph-based tasks as a testbed for studying multi-step reasoning with CoTs [28, 30, 27].

**Test-time scaling**  Extensive work focused on scaling inference-time compute optimally [12, 1, 56, 57], in search of inference-time scaling laws [2, 58, 59, 60]. A line of work has focused on studying optimal sequential scaling [7, 21, 61, 62] by examining the role of CoT length [63, 64, 65, 66]. The benefits of learning to search [67, 68, 69, 70] and problem-solving strategies like backtracking and self-correction [71, 19, 72] by scaling the CoT length have also been demonstrated [73, 74, 75], as well as the limits of these approaches [76, 14]. Another line of work has studied parallel scaling [15, 13]

---

**Prompt format**

```
Given the following list of undirected edges in a graph (with nodes labeled 0 through 33), is node
0 in the same component as 10 or as 27?  (it is connected to exactly one of the two) Think step by
step.
(11, 12), (23, 24), (6, 7), (25, 17), (4, 5), (27, 28), (9, 10), (2, 16), (13, 14), (2, 3), (5, 6),
(18, 17), (10, 11), (3, 4), (31, 32), (18, 19), (19, 33), (30, 31), (20, 21), (2, 9), (24, 25),
(15, 16), (12, 13), (7, 8), (19, 20), (1, 2), (32, 33), (29, 30), (14, 15), (28, 29), (1, 0), (8,
0), (21, 22), (19, 26), (22, 23), (26, 27)
```

---

Figure 12: Example prompt from the LLM experiments. The prompt includes basic instructions for the task, along with the recommendation to think step by step (to avoid the model responding immediately with a guess, and then spending the rest of the chain of thought trying to justify it).

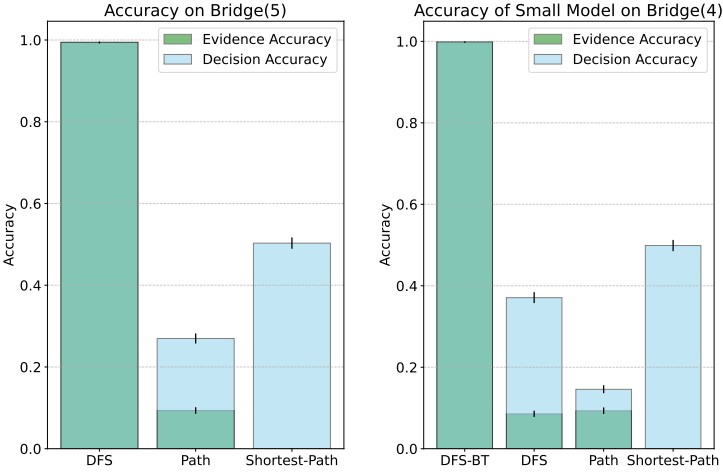

Figure 13: Decision and evidence accuracy of (left) models trained on CoT strategies for `Bridge(5)` task, and (right) models with 2 hidden layers trained on CoT strategies, including DFS-BT and DFS, for `Bridge(5)` task. Error bars represent 95% binomial confidence intervals.

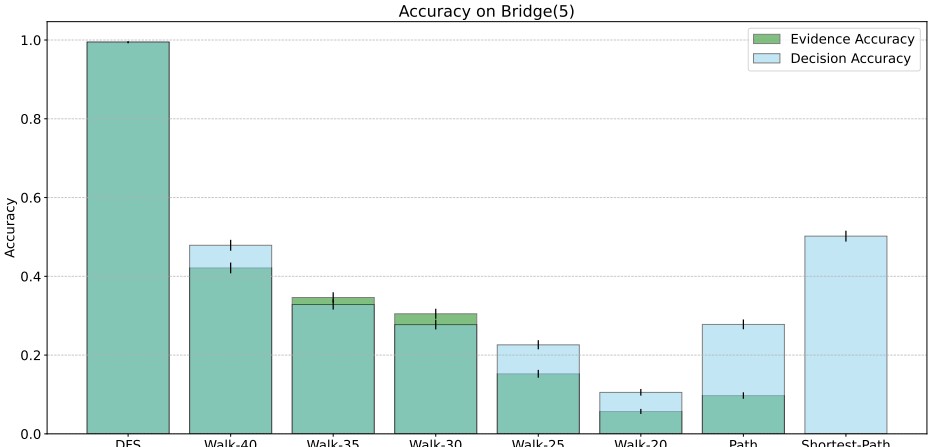

Figure 14: Decision and evidence accuracy of models trained on Walk CoT strategies for `Bridge(5)` task. Error bars represent 95% binomial confidence intervals.

by examining the behavior of majority voting or a best-of-$n$ method over a diverse set of responses generated in parallel [77, 78]. Finally, the role of reinforcement learning [79, 80, 81] in advancing reasoning by improving the CoT quality and scaling it naturally [6, 20, 22, 39, 40] has been explored.

