# OpenReview forum: "Let Me Think! A Long Chain of Thought Can Be Worth Exponentially Many Short Ones"
_NeurIPS.cc/2025/Conference — NeurIPS 2025 poster_

### Official Review · Reviewer_2f9M · 2025-06-11

**Clarity:** 3
**Significance:** 3
**Originality:** 3
**Rating:** 5
**Confidence:** 4

**Summary:**

This paper investigates how to optimally allocate computational resources during reasoning tasks. Through both theoretical analysis and empirical experiments, the authors demonstrate that sequential scaling (e.g., using longer Chain-of-Thought, or CoT) can offer exponential advantages over parallel scaling (e.g., majority voting or best-of-N strategies) in certain graph connectivity problems.
The main contribution of the work is the introduction of a reasoning task where slightly reducing the sequential budget requires a significant increase in the parallel budget to maintain the same level of accuracy.
The authors also explore the behavior of transformers trained with reinforcement learning (RL) on this task, observing that the length of the generated CoT increases gradually during training.

**Questions:**

See Weaknesses

**Ethical Concerns:**

["NO or VERY MINOR ethics concerns only"]

**Final Justification:**

The authors have addressed most of my concerns. So I decide to raise my score to 4.

**Limitations:**

Yes

**Quality:**

2

**Strengths And Weaknesses:**

**Strengths**

+ The paper not only presents a theoretical analysis of the differences between sequential and parallel scaling in reasoning tasks but also validates these predictions through experiments.
+ The authors design a graph connectivity-based reasoning task that is both challenging and capable of simulating multi-step reasoning, offering an ideal testbed for studying the relative advantages of sequential scaling.

**Weaknesses**

+ This paper **mainly focuses on a graph connectivity task,** which is relatively simple and may not reflect the complexity of real-world reasoning scenarios. I believe deeper analysis should be conducted on more complex reasoning domains such as mathematical problem-solving (e.g., Math, AIME) or code generation (e.g. LivecodeBench) to better validate the generalizability of the findings.

+ The claim that reinforcement learning leads to an increase in CoT length is questionable. In practical reasoning-enhancement training settings, such as those involving mathematics or coding, they often use supervised fine-tuning to extend the length of reasoning chains first, followed by RL to achieve distribution shifts. In this process, RL does not necessarily result in longer CoTs.

+ The vertex query model abstracts graph access via an oracle $N_G$, limiting algorithms to querying only the neighbors of specific vertices. While useful for theoretical analysis, this abstraction may overlook the complex interactions and capabilities of models when processing graph-structured data.

+ The paper **mainly introduced an observed phenomenon**—that increasing the model’s generation length significantly boosts reasoning performance. However, similar conclusions have been reported in the previous paper. For instance, methods such as process reward modeling or MCTS for parallel scaling are being increasingly abandoned due to high noise, with recent trends focusing instead on extending the length of reasoning chains with RL.

+ In addition, I recommend that the paper should propose a new solution under specific scenarios, that is, **how to balance sequential and parallel scaling under constrained inference resources**. Addressing these points with concrete experiments would not only enhance the robustness of the findings but also increase the overall impact of the work.

---

> ### Author Rebuttal · Authors · 2025-07-29
>
> Thank you for the constructive feedback and thoughtful comments. We are glad that you found our graph connectivity task an ideal testbed for theoretical study of the tradeoffs between sequential and parallel scaling for reasoning models and validating it through controlled experiments. Below we address your concerns.
>
> > This paper mainly focuses on a graph connectivity task, which is relatively simple and may not reflect the complexity of real-world reasoning scenarios.
>
> Our goal in this paper is to make a general and fundamental claim about tradeoffs between parallel and sequential scaling. Our understanding of these tradeoffs is at its infancy. Multiple papers in the literature have seemingly contradictory claims: some show large benefits from sequential scaling [1], while other papers claim that parallel scaling alone is sufficient [2].
>
> Our goal is to add clarity to the literature, by providing a controllable and analyzable task in which sequential scaling (theoretically & empirically) cannot be efficiently replaced by parallel scaling. Thus, our results yield the general conclusion that the parallel scaling recipes of papers such as [2, 3] cannot work for all types of problems.
>
> Furthermore, we propose the graph connectivity task because it is very natural and captures multi-step reasoning ability, which seems to be a key aspect of more complex reasoning tasks such as math problems. Reasoning on graphs has been considered in the literature as an ideal abstraction of complex reasoning tasks (see [4,5,6,7] for examples), which also isolates the reasoning ability from memorization. Motivated by that, we have designed the bridge graph connectivity task, to capture key aspects of multi-step reasoning.
>
> To be clear, our claim is not that sequential scaling is always better than parallel scaling for all tasks. Rather, it is that there is a natural class of tasks where it is (empirically and theoretically) better. **We are the first to formally show such a gap for transformers in a natural setting that is a building block for more advanced multi-step reasoning tasks.** This is a conceptual contribution that we believe is helpful for thinking about more complex tasks. We will edit the introduction and abstract to be more clear on this point so as to avoid confusion.
>
>
>
> > I believe deeper analysis should be conducted on more complex reasoning domains such as mathematical problem-solving (e.g., Math, AIME) or code generation (e.g. LivecodeBench) to better validate the generalizability of the findings.
>
> We have run some evaluations of s1 [1] on GPQA Diamond, and compared sequential and parallel scaling in that setting (we cannot include any plots due to the removal of the option to share a pdf). These experiments show sequential vs. parallel tradeoffs qualitatively similar to our experiments on the graph connectivity task. As argued above, we believe that these experiments are not core to our message and that the paper stands on its own without them. Nevertheless, we will include these experiments in the camera ready paper, since they support our claim that graph connectivity is a helpful benchmark to consider.
>
> > The paper mainly introduced an observed phenomenon—that increasing the model’s generation length significantly boosts reasoning performance.
>
> It is indeed known that increasing sequential scale can improve performance, see e.g. [1]. However, this is not the point of our paper.
>
> Instead, our paper studies the tradeoff between parallel and sequential scaling, which is much less understood. Some works argue that parallel scaling alone is enough to get very good performance [2, 3]. Our paper demonstrates that a tradeoff exists between parallel scaling and sequential scaling for a certain natural class of graph connectivity tasks. Thus, our paper pushes back against [2, 3], showcasing a simple setting where parallel scaling alone is not enough.
>
>
> > The claim that reinforcement learning leads to an increase in CoT length is questionable.
>
> We agree that RL is not the only way to enable long CoT reasoning in practice and supervised fine-tuning first was the standard approach prior to the release of DeepSeek-R1 [8, 9]. However, DeepSeek-R1-Zero [10] and its replications [11, 12] showed that even scaling up RL on a base model without supervised fine-tuning can lead to increase in the model's CoT length and accuracy (see Figures 2 and 3 in [10], as well as Figure 1 and Section 4 in [11]). Our study gives one plausible explanation for this phenomena from an expressivity lens of view, demonstrating a clear example of how RL can improve model's accuracy by reinforcing long CoTs the model is expressive enough to compute.
>
> > The vertex query model abstracts graph access via an oracle, limiting algorithms to querying only the neighbors of specific vertices. While useful for theoretical analysis, this abstraction may overlook the complex interactions and capabilities of models when processing graph-structured data.
>
> The vertex query model is an abstraction that we empirically certify works extremely well in our from-scratch training experiments (see Figure 4). If transformers had significantly more capabilities in this setting, one would expect them to be able to correctly learn the shortest path, but empirically the model can’t learn it non-trivially better than the vertex query model would suggest. Additionally, the results implied by the vertex query model hold well even with our LLM experiments.
> The results with the vertex query model are complementary to Theorem 1 (which does not use the vertex query model). Theorem 1 tells us that the separation qualitatively holds, and the Vertex Query model allows us to get a much more fine-grained result in the setting of graph connectivity (albeit with more assumptions).
>
> > The paper should propose a new solution under specific scenarios, that is, how to balance sequential and parallel scaling under constrained inference resources.
>
> We have done a grid search over combinations of sequential and parallel scaling (see Figures 1 and 5), showing how different combinations perform. In practice, the optimal tradeoff depends on the specific implementations, hardware constraints, and the setting. To make a fair comparison, one would have to optimize both techniques separately. That being said, we would be happy to add wallclock time comparisons for our experiments to the camera ready version of the paper.
> Theoretically, sequential scaling should be quadratic (with transformer architecture) in the CoT budget, while parallel should be linear. Since we show an exponential gap, this means that sequential scaling will always be favored in the limit.
> We are identifying a natural setting where sequential scaling can not be efficiently replaced by parallel scaling, however different specific settings can have different tradeoffs, and in such settings the optimal balance would be different. We believe that future works can use this work as a foundation to investigate the optimal tradeoff for specific settings, constrained to specific computational resources.
>
> Thank you again for your careful reading of the paper and thoughtful suggestions. We hope that we have addressed your questions sufficiently.
>
> [1] Muennighoff, Niklas, et al. "s1: Simple test-time scaling." arXiv preprint arXiv:2501.19393 (2025).
>
> [2] Ma, Wenjie, et al. "Reasoning models can be effective without thinking." arXiv preprint arXiv:2504.09858 (2025).
>
> [3] Brown, Bradley, et al. "Large language monkeys: Scaling inference compute with repeated sampling." arXiv preprint arXiv:2407.21787 (2024).
>
> [4] Abbe, Emmanuel, et al. "How far can transformers reason? the globality barrier and inductive scratchpad." Advances in Neural Information Processing Systems 37 (2024): 27850-27895.
>
> [5] Sanford, Clayton, et al. "Understanding transformer reasoning capabilities via graph algorithms." Advances in Neural Information Processing Systems 37 (2024): 78320-78370.
>
> [6] Xu, Keyulu, et al. "What can neural networks reason about?." arXiv preprint arXiv:1905.13211 (2019).
>
> [7] Kim, Juno, et al. "Metastable dynamics of chain-of-thought reasoning: Provable benefits of search, rl and distillation." arXiv preprint arXiv:2502.01694 (2025).
>
> [8] "Demystifying Long Chain-of-Thought Reasoning in LLMs", Yeo et al. 2025
>
> [9] "Kimi k1.5: Scaling Reinforcement Learning with LLMs", Kimi Team 2025
>
> [10] "DeepSeek-R1: Incentivizing Reasoning Capability in LLMs via Reinforcement Learning", DeepSeek-AI 2025
>
> [11] "SimpleRL-Zoo: Investigating and Taming Zero Reinforcement Learning for Open Base Models in the Wild", Zeng et al. 2025
>
> [12] "TinyZero", Pan et al. 2025

---

> > ### Author Response · Authors · 2025-08-05
> >
> > We hope our review addressed your concerns to your satisfaction, sufficiently for you to consider raising your score! We are happy to hear any remaining concerns, confusions, or feedback about the paper.

---

> ### Comment · Reviewer_2f9M · 2025-08-06
>
> Thank you for your response. Considering that you do not conduct experiments on many common real-world reasoning scenarios (i.e. AIME 24, AIME 25), we think that this paper has limitations.
> So I decide to maintain my score.

---

> > ### Author Response · Authors · 2025-08-06
> >
> > We have run some evaluations of s1 [1] on GPQA Diamond, and compared sequential and parallel scaling in that setting (we cannot include any plots due to the removal of the option to share a pdf). These experiments show sequential vs. parallel tradeoffs qualitatively similar to our experiments on the graph connectivity task. As argued above, we believe that these experiments are not core to our message and that the paper stands on its own without them. Nevertheless, we will include these experiments in the camera ready paper, since they support our claim that graph connectivity is a helpful benchmark to consider.
> >
> > If you believe that AIME experiments are essential and would be sufficient to raise your score, we can run them in the next 2 days.

---

> > ### Author Response · Authors · 2025-08-08
> >
> > In response to your valuable feedback, and to further assess the generalizability of our conclusions to more complex reasoning tasks, we conducted additional experiments on AIME-2024 and posted the results as an official comment. We hope these results address your remaining concerns and encourage you to consider raising your score.

---

> > > ### Comment · Reviewer_2f9M · 2025-08-08
> > >
> > > Considering that this article has too few experiments in real scenarios and has not yet run the AIME dataset, I decided to lower my score.

---

> ### Author Response · Authors · 2025-08-08
>
> Dear Reviewer 2f9M,
> Perhaps you did not see our "Official comment" since we posted it as a response to all reviewers. We have pasted its contents below here so that you can more easily see it. The experiments that we have run on AIME support our theoretical results.
>
> --
>
> In response to the reviewers’ feedback, we conducted additional experiments with the s1‑32B model [1] on AIME‑2024. For parallel scaling, we sample with temperature 1.0 and aggregate by majority vote over final answers. For sequential scaling, we limit the model’s thinking‑token budget and force a final answer once the limit is reached. In the ‘wait’ variant [1], we ignore the model’s first output and append the ‘wait’ token to induce further reasoning before the final answer. The experiments used ≈24 H200 GPU‑hours. Since PDF attachments are not supported here, we present the results in the table below.
>
> | Parallel (maj@k) / Sequential (token) | 500  | 1k   | 2k   | 4k   | 8k   | wait |
> |-------------------------------------------------------------|------|------|------|------|------|--------------|
> | 1                                                           | 0.067 | 0.133 | 0.200 | 0.333 | 0.300 | 0.433        |
> | 2                                                           | 0.067 | 0.133 | 0.200 | 0.333 | 0.300 | 0.433        |
> | 4                                                           | 0.100 | 0.133 | 0.233 | 0.433 | 0.433 | 0.500        |
> | 8                                                           | 0.067 | 0.133 | 0.300 | 0.433 | 0.467 | 0.533        |
> | 16                                                          | 0.067 | 0.167 | 0.333 | 0.433 | 0.500 | 0.567        |
> | 32                                                          | 0.067 | 0.133 | 0.400 | 0.433 | 0.533 | 0.600        |
> | 64                                                          | 0.067 | 0.133 | 0.400 | 0.433 | 0.533 | 0.567        |
> | Avg. #Thinking Tokens                                        | 500  | 1000  | 2000  | 3998  | 5092  | 5522         |
>
>
>
> The results show that sequential scaling can not be efficiently replaced by parallel scaling for this mathematical task, supporting the generalizability of our findings to real-world scenarios. While quantifying the exact trade-off between them for complex mathematical problems such as this is beyond the scope of our fundamental study, we observe that the results confirm our conclusion that sequential scaling is necessary, and challenge claims from other works [2] that it can be entirely replaced by parallel scaling.
>
>
> To further examine this observation, we break down the results by individual question (out of the 30 total). The table below reports the number of correct responses out of 64 attempts for each question.
>
> | Sequential (token budget) / Question ID |  1 |  2 |  3 |  4 |  5 |  6 |  7 |  8 |  9 | 10 | 11 | 12 | 13 | 14 | 15 | 16 | 17 | 18 | 19 | 20 | 21 | 22 | 23 | 24 | 25 | 26 | 27 | 28 | 29 | 30 |
> |---------------------------|----|----|----|----|----|----|----|----|----|----|----|----|----|----|----|----|----|----|----|----|----|----|----|----|----|----|----|----|----|----|
> | 500                       |  0 |  0 |  0 |  0 |  1 |  0 |  0 | 36 | 12 |  0 |  0 |  0 | 13 |  0 |  0 |  0 |  0 |  2 |  3 |  0 |  0 |  0 |  0 |  0 |  0 |  0 |  0 |  0 |  0 |  0 |
> | 1k                        |  0 |  1 |  0 |  0 |  6 |  0 |  0 | 55 | 36 |  4 |  0 |  4 | 48 |  0 |  0 |  2 |  0 |  1 |  1 |  0 |  0 |  1 |  0 | 27 |  0 |  0 |  0 |  0 |  0 |  0 |
> | 2k                        | 56 |  1 |  0 |  0 | 20 |  0 |  0 | 63 | 23 | 55 |  0 | 29 | 47 |  0 |  1 |  9 |  1 |  1 |  4 | 12 |  0 |  0 | 12 | 34 | 57 |  0 | 15 |  0 |  0 |  0 |
> | 4k                        | 58 |  1 |  0 |  0 | 16 |  0 |  2 | 63 | 22 | 60 |  2 | 43 | 58 |  1 |  4 | 27 |  0 |  3 |  3 | 31 |  0 |  0 | 36 | 50 | 62 |  0 | 33 |  5 |  0 |  0 |
> | 8k                        | 62 |  2 |  0 |  0 |  5 |  2 | 24 | 64 | 28 | 63 |  1 | 49 | 58 |  9 | 16 | 40 |  2 |  2 | 16 | 33 |  2 |  0 | 29 | 51 | 63 |  2 | 36 |  8 |  0 |  0 |
> | wait                      | 61 |  0 |  0 |  0 |  5 |  0 | 35 | 64 | 22 | 62 |  6 | 48 | 60 | 12 | 19 | 44 |  7 |  6 | 19 | 34 |  0 |  0 | 31 | 50 | 60 |  2 | 31 |  7 |  0 |  0 |
>
>
>
> One notable takeaway is that there exist questions (e.g. 1, 10, 25) for which parallel scaling at low token budgets will never succeed, but a single shot with a higher token budget is very likely to succeed. This shows the necessity of sequential scaling for achieving the highest possible score on this task. We hope this further clarifies the generalizability of the insights offered by our work.
>
> [1] Muennighoff, Niklas, et al. "s1: Simple test-time scaling." arXiv preprint arXiv:2501.19393 (2025).
>
> [2] Ma, Wenjie, et al. "Reasoning models can be effective without thinking." arXiv preprint arXiv:2504.09858 (2025).

---

### Official Review · Reviewer_GJ4H · 2025-06-26

**Clarity:** 3
**Significance:** 2
**Originality:** 3
**Rating:** 5
**Confidence:** 4

**Summary:**

This paper investigates the fundamental tradeoff between sequential scaling (longer chains of thought) and parallel scaling (multiple short chains with aggregation) in large language model reasoning. The authors demonstrate through both theoretical analysis and empirical validation that sequential scaling can offer exponential advantages over parallel scaling on graph connectivity problems. They introduce a "bridge graph" task based on graph connectivity and show that small decreases in sequential scale require large increases in parallel scale to maintain performance.

**Questions:**

Overall, this paper makes solid theoretical and empirical contributions to an important problem in current LLM research. However, the limited scope to graph connectivity problems and reliance on unproven complexity-theoretic assumptions prevent it from being a clear accept. The work provides valuable insights into inference-time scaling but needs broader validation to have significant impact.  If the authors can adequately address the concerns raised in the Limitations/Weaknesses part, I would be willing to increase my score.


**Additional Questions**

1. Limited Empirical Experiment Scope: Though adapting the theoretical analysis could be tough, the authors can easily conduct empirical analysis on sequential scaling and parallel scaling for other LLM reasoning tasks. This will help to better validate the effectiveness of the authors claims in more diverse settings.

**Ethical Concerns:**

["NO or VERY MINOR ethics concerns only"]

**Final Justification:**

The authors have addressed my concerns, so I am inclined to raise my score.

**Limitations:**

Yes

**Quality:**

3

**Strengths And Weaknesses:**

**Strengths and Contributions**

1.	Novel theoretical framework: The paper provides rigorous theoretical analysis using transformer expressivity limitations on graph reasoning tasks.

2.	Sufficient empirical validation: Various models are assessed and consistently validate the theoretical predictions.

3.	Practical insights: The work addresses a fundamental question in inference-time scaling that is highly relevant to current LLM development.

**Limitations / Weaknesses**

1.	Limited scope of tasks: The results are demonstrated only on graph connectivity problems. While the authors acknowledge this limitation, it's unclear how broadly these findings generalize to other reasoning domains.

2.	Noisy Empirical Results: In some plots of empirical results, including the right plot of Figure 1 and the two plots in Figure 5, the results are not visibly exponential. The curve type of the sequential-parallel tradeoff needs to be further determined.

3.	Limited exploration of hybrid approaches: The paper focuses primarily on pure sequential vs. pure parallel scaling, with less investigation of optimal combinations of both approaches.

---

> ### Author Rebuttal · Authors · 2025-07-31
>
> Thank you for your thoughtful comments and helpful feedback! We are glad that you found our theoretical framework and its empirical validation novel and solid to address the fundamental tradeoffs in inference-time scaling and giving practical insights.
>
> > The results are demonstrated only on graph connectivity problems. While the authors acknowledge this limitation, it's unclear how broadly these findings generalize to other reasoning domains
>
> Our goal in this paper is to make a general and fundamental claim about tradeoffs between parallel and sequential scaling. Our understanding of these tradeoffs is at its infancy. Multiple papers in the literature have seemingly contradictory claims: some show large benefits from sequential scaling [1], while other papers claim that parallel scaling alone is sufficient [2].
>
> Our goal is to add clarity to the literature, by providing a controllable and analyzable task in which sequential scaling (theoretically & empirically) cannot be efficiently replaced by parallel scaling. Thus, our results yield the general conclusion that the parallel scaling recipes of papers such as [2, 3] cannot work for all types of problems.
>
> Furthermore, we propose the graph connectivity task because it is very natural and captures multi-step reasoning ability, which seems to be a key aspect of more complex reasoning tasks such as math problems. Reasoning on graphs has been considered in the literature as an ideal abstraction of complex reasoning tasks (see [4,5,6,7] for examples), which also isolates the reasoning ability from memorization. Motivated by that, we have designed the bridge graph connectivity task, to capture key aspects of multi-step reasoning.
>
> To be clear, our claim is not that sequential scaling is always better than parallel scaling for all tasks. Rather, it is that there is a natural class of tasks where it is (empirically and theoretically) better. We are the first to formally show such a gap for transformers in a natural setting that is a building block for more advanced multi-step reasoning tasks. This is a conceptual contribution that we believe is helpful for thinking about more complex tasks. We will edit the introduction and abstract to be more clear on this point so as to avoid confusion.
>
>
>
> Finally, we have run some evaluations of s1 [1] on GPQA Diamond, and compared sequential and parallel scaling in that setting (we cannot include any plots due to the removal of the option to share a pdf). These experiments show sequential vs. parallel tradeoffs qualitatively similar to our experiments on the graph connectivity task. As argued above, we believe that these experiments are not core to our message and that the paper stands on its own without them. Nevertheless, we will include these experiments in the camera ready paper, since they support our claim that graph connectivity is a helpful benchmark to consider.
>
> >  In some plots of empirical results, including the right plot of Figure 1 and the two plots in Figure 5, the results are not visibly exponential.
>
> Our results only show that the tradeoff is at *least* exponential. In those plots, the contours of equal loss appear to possibly grow faster than exponential, so do not contradict our results. It is not too surprising that the LLM experiments don’t perfectly match the smaller scale ones, since LLMs are not trained on this specific task, so will spend tokens figuring out what solution they will try, and won’t make assumptions about the structure of the graph and therefore are likely to attempt suboptimal solutions. We will make sure to add more discussion and concrete examples of this observation to the paper.
>
> > The paper focuses primarily on pure sequential vs. pure parallel scaling, with less investigation of optimal combinations of both approaches.
>
> We have done a grid search over combinations of sequential and parallel scaling (see Figures 1 and 5), showing how different combinations perform. In practice, the optimal tradeoff depends on the specific implementations, hardware constraints, and the setting. We believe that future works can use this work as a foundation to investigate the optimal combinations for specific settings.
>
> > Reliance on unproven complexity-theoretic assumptions.
>
> The assumption that $\textrm{TC}^0 \neq \textrm{L}$ is a standard assumption (see [8,9]) commonly held by complexity theorists (similarly to the conjecture that $\textrm{P} \neq \textrm{NP}$). We explicitly mention this assumption whenever we use it, and also have theoretical results (see Theorem 1) that do not rely on this assumption.
>
> Thank you again for your careful reading of the paper and thoughtful suggestions. We hope that we have addressed your questions sufficiently.
>
> [1] Muennighoff, Niklas, et al. "s1: Simple test-time scaling." arXiv preprint arXiv:2501.19393 (2025).
>
> [2] Ma, Wenjie, et al. "Reasoning models can be effective without thinking." arXiv preprint arXiv:2504.09858 (2025).
>
> [3] Brown, Bradley, et al. "Large language monkeys: Scaling inference compute with repeated sampling." arXiv preprint arXiv:2407.21787 (2024).
>
> [4] Abbe, Emmanuel, et al. "How far can transformers reason? the globality barrier and inductive scratchpad." Advances in Neural Information Processing Systems 37 (2024): 27850-27895.
>
> [5] Sanford, Clayton, et al. "Understanding transformer reasoning capabilities via graph algorithms." Advances in Neural Information Processing Systems 37 (2024): 78320-78370.
>
> [6] Xu, Keyulu, et al. "What can neural networks reason about?." arXiv preprint arXiv:1905.13211 (2019).
>
> [7] Kim, Juno, et al. "Metastable dynamics of chain-of-thought reasoning: Provable benefits of search, rl and distillation." arXiv preprint arXiv:2502.01694 (2025).
>
> [8] Feng, Guhao, et al. "Towards revealing the mystery behind chain of thought: a theoretical perspective." Advances in Neural Information Processing Systems 36 (2023): 70757-70798.
>
> [9] Merrill, William, and Ashish Sabharwal. "The expressive power of transformers with chain of thought." arXiv preprint arXiv:2310.07923 (2023).

---

> > ### Comment · Reviewer_GJ4H · 2025-08-05
> >
> > Since the authors have addressed my concerns, I am keeping my original borderline accept rating.

---

> > > ### Author Response · Authors · 2025-08-05
> > >
> > > Thanks for your response! If we have adequately addressed the concerns raised in the Limitations/Weaknesses part of your review, we hope you will consider increasing your score (as you mentioned you would in your review).

---

> > > > ### Comment · Reviewer_GJ4H · 2025-08-06
> > > >
> > > > The authors have adequately addressed most concerns, and the work makes a solid contribution to understanding inference-time scaling, which is highly relevant given current trends in reasoning models. I will increase my score to 5.
> > > >
> > > > However, I want to strongly emphasize a crucial limitation:
> > > > To support a general claim for the community to choose between parallel and sequential scaling still requires more experiments than those presented. The authors must include more experiments across diverse reasoning domains in the camera-ready version to sufficiently support this broader claim.
> > > > While the theoretical analysis on graph connectivity is rigorous, the generalizability to other reasoning tasks remains the paper's biggest weakness. The promised GPQA Diamond experiments are a start, but the community needs evidence across mathematical reasoning (MATH, GSM8K, SVAMP), code generation, and other reasoning benchmarks. The authors should view the camera-ready version as an opportunity to significantly expand their empirical scope.

---

> ### Author Response · Authors · 2025-08-08
>
> Thank you for your valuable feedback. To further assess the generalizability of our conclusions to more complex reasoning tasks, such as mathematical reasoning, we conducted additional experiments on AIME-2024 and posted the results as an official comment. We are also running experiments with other models, such as Qwen3-32B, and other mathematical tasks, such as MATH-500 for the camera-ready version, to make the empirical study more comprehensive, and further clarify the generalizability of the insights offered by our work.

---

### Official Review · Reviewer_WjKT · 2025-06-30

**Clarity:** 3
**Significance:** 2
**Originality:** 3
**Rating:** 3
**Confidence:** 3

**Summary:**

This paper focuses on inference-time computation and compares parallel scaling and sequential scaling to quantify the trade-offs between the two for reasoning tasks. The authors study the graph connectivity task and provide both theoretical analysis and empirical evidence to demonstrate that sequential scaling outperforms parallel scaling.

**Questions:**

1. The paper needs a more detailed analysis of why sequential scaling performs better than parallel scaling. For example, does longer generated text in sequential scaling contain more diverse or richer information, whereas parallel generation may suffer from repetition across outputs? Providing concrete examples or case studies would help readers intuitively understand the issue.
2. More details about the training and evaluation data are needed—especially the nature of the context provided and the size of the test set.
3. What specific strategy is employed to ensure that longer chains of thought are meaningful and not simply verbose or noisy?
4. What is the comparison in actual time cost between sequential and parallel scaling under a fixed token budget?
5. Does model size affect the performance gap between sequential and parallel scaling?

**Ethical Concerns:**

["NO or VERY MINOR ethics concerns only"]

**Final Justification:**

The author's responses provide clarifications. While some of my concerns are resolved, I still concern about the novelty of the technique, as well as some experimental details. I will therefore maintain my original evaluation.

**Limitations:**

yes

**Quality:**

2

**Strengths And Weaknesses:**

Strengths：
1. The paper clearly defines the problem of graph connectivity, investigates it thoroughly, and supports its claims with both theoretical analysis and empirical evidence, demonstrating that sequential scaling outperforms parallel scaling.
2. The integration of reinforcement learning into the analysis adds technical depth and contributes to the novelty of the work.
3. The paper is well-organized, with a logical and coherent structure that enhances clarity and readability.

Weakness：
1. The graph connectivity task is inherently sequential in nature. Could the observed performance gains be simply due to this sequential characteristic? In other words, is the task biased in favor of sequential scaling, and does that limit the generality of the conclusion?
2. How does the proposed approach perform on other reasoning tasks, such as mathematical problem solving, which may involve different forms of reasoning?
3. Since the paper focuses primarily on the graph connectivity task, it may be helpful to clarify this focus more explicitly in the title or abstract, to better manage reader expectations and highlight the scope of the study.

---

> ### Author Rebuttal · Authors · 2025-07-31
>
> Thank you for your detailed response and thoughtful comments. We are glad that you found our paper well-organized and coherent, providing both theoretical analysis and empirical evidence that support our claims. Below we address your concerns.
>
> > The graph connectivity task is inherently sequential in nature. Could the observed performance gains be simply due to this sequential characteristic? In other words, is the task biased in favor of sequential scaling, and does that limit the generality of the conclusion?
>
> Our understanding of the tradeoffs between parallel and sequential scaling is at its infancy. Multiple papers in the literature have seemingly contradictory claims: some show large benefits from sequential scaling [1], while other papers claim that parallel scaling alone is sufficient [2].
>
> Our goal is to add clarity to the literature, by providing a task in which sequential scaling (theoretically & empirically) cannot be efficiently replaced by parallel scaling. Thus, **our results yield the general conclusion that the parallel scaling recipes of papers such as [2, 3] cannot work for all types of problems.** We propose the graph connectivity task because it is very natural and captures multi-step reasoning ability.
>
> To be clear, our claim is not that sequential scaling is always better than parallel scaling for all tasks. Rather, it is that there is a natural class of tasks where it is (empirically and theoretically) better – and this class of tasks seems to capture key aspects of multi-step reasoning. We will edit the introduction and abstract to be more clear on this point so as to avoid confusion.
>
> Finally, we have run some evaluations of s1 [1] on GPQA Diamond, and compared sequential and parallel scaling in that setting (we cannot include any plots due to the removal of the option to share a pdf). These experiments show sequential vs. parallel tradeoffs qualitatively similar to our experiments on the graph connectivity task. As argued above, we believe that these experiments are not core to our message and that the paper stands on its own without them. Nevertheless, we will include these experiments in the camera ready paper, since they support our claim that graph connectivity is a helpful benchmark to consider.
>
> > How does the proposed approach perform on other reasoning tasks, such as mathematical problem solving, which may involve different forms of reasoning?
>
> Our paper’s goal is to improve our theoretical understanding of CoT scaling, instead of proposing a new approach. We are motivated by the empirical literature around sequential and parallel scaling [1, 2] and understanding the fundamental tradeoff between them for more complex tasks such as mathematical problem solving.
> We agree that different specific tasks may have different tradeoffs and we do not claim that sequential scaling is always better than parallel scaling (see our response to the question above). Our study shows the non-trivial necessity of sequential scaling for a general multi-step reasoning problem, both theoretically (based on expressivity results), and empirically (through comprehensive controlled experiments). **We are the first to formally show such a gap for transformers in a natural setting that is a building block for more advanced multi-step reasoning tasks.** This is a conceptual contribution that we believe is helpful for thinking about more complex tasks.
>
> > It may be helpful to clarify the focus on the graph connectivity task more explicitly in the title or abstract.
>
> We have mentioned in the abstract that our paper demonstrates the existence of settings where sequential scaling is necessary based on graph connectivity problems, but we will emphasize it more in the revision.
>
> > The paper needs a more detailed analysis of why sequential scaling performs better than parallel scaling. For example, does longer generated text in sequential scaling contain more diverse or richer information, whereas parallel generation may suffer from repetition across outputs?
>
> Thank you for the helpful recommendation. We provide a theoretical explanation in the submission’s Lines 161-165: A limited depth model can only check connectivity for (relatively) nearby vertices, so to find the correct path it has to explore mostly “blind” until it stumbles upon the solution. In Lines 245-258, we corroborate this explanation by investigating how models trained on short CoT behave: the model has the same rate of success as if it were exploring blindly.
>
> Your comment helped us see that this is hard to follow because our theoretical arguments and empirical corroboration are in separate sections. We will unite these two paragraphs in the revision and add more detail to make it easier to understand.
>
>
> > More details about the training and evaluation data, the nature of the context provided and the size of the test set.
>
> For the from-scratch training experiment, we include the details in sections 4.1, 4.2, and 4.3. An example prompt is in Figure 2. The test set size is 5,000, thanks for pointing out, it will be added to help supplement the already provided confidence intervals. For the LLM experiments, all of the details are in C.4, including an example prompt (Figure 10), size of the test set, and details of the CoT context length. For full transparency of our experiments and to aid with reproducing our results, our code was included in the supplementary details.
>
> > What specific strategy is employed to ensure that longer chains of thought are meaningful and not simply verbose or noisy?
>
> For experiments on pretrained reasoning LLMs, the accuracy improves with sequential scale. For experiments on transformers trained from scratch, our training data consists of longer chains of thought.
>
> For evaluation in the from-scratch training experiments, in addition to our decision criterion which checks the final result, we have an evidence criterion which verifies whether the chain-of-thought generated by the model contains a valid trace as a solution. We report the evidence accuracy metric based on this criterion to make sure the chains of thought are meaningful.
>
> > What is the comparison in actual time cost between sequential and parallel scaling under a fixed token budget?
>
> In practice, this depends on the specific implementations, hardware constraints, and the setting. To make a fair comparison, one would have to optimize both techniques separately. That being said, we would be happy to add wallclock time comparisons for our experiments to the camera ready version of the paper.
> Theoretically, sequential scaling should be quadratic in the CoT budget (in the limit), while parallel should be linear. Since we show an exponential gap, this means that sequential scaling will always be favored in the limit.
>
> > Does model size affect the performance gap between sequential and parallel scaling?
>
> Our experiments with small transformers and LLMs show that the overall trends look similar across vastly different scales (though since there are a lot of other differences between the experiments, we would caution against quantitative take-aways). Theoretically, for any constant depth model, there will always be an exponential gap, but for deeper models, they are expressive enough to do more in one step, and so may need less sequential scaling.
>
> Thank you again for your careful reading of the paper and thoughtful suggestions. We hope that we have addressed your questions sufficiently.
>
> [1] Muennighoff, Niklas, et al. "s1: Simple test-time scaling." arXiv preprint arXiv:2501.19393 (2025).
>
> [2] Ma, Wenjie, et al. "Reasoning models can be effective without thinking." arXiv preprint arXiv:2504.09858 (2025).
>
> [3] Brown, Bradley, et al. "Large language monkeys: Scaling inference compute with repeated sampling." arXiv preprint arXiv:2407.21787 (2024).
>
> [4] Abbe, Emmanuel, et al. "How far can transformers reason? the globality barrier and inductive scratchpad." Advances in Neural Information Processing Systems 37 (2024): 27850-27895.
>
> [5] Sanford, Clayton, et al. "Understanding transformer reasoning capabilities via graph algorithms." Advances in Neural Information Processing Systems 37 (2024): 78320-78370.
>
> [6] Xu, Keyulu, et al. "What can neural networks reason about?." arXiv preprint arXiv:1905.13211 (2019).
>
> [7] Kim, Juno, et al. "Metastable dynamics of chain-of-thought reasoning: Provable benefits of search, rl and distillation." arXiv preprint arXiv:2502.01694 (2025).

---

> ### Author Response · Authors · 2025-08-05
>
> We hope our review addressed your concerns to your satisfaction, sufficiently for you to consider raising your score! We are happy to hear any remaining concerns, confusions, or feedback about the paper.

---

> > ### Comment · Reviewer_WjKT · 2025-08-06
> >
> > Thank you for the clarifications and thoughtful response. While some of my concerns are resolved, I still concern about the novelty of the technique, as well as some experimental details. I will therefore maintain my original evaluation.

---

> > > ### Author Response · Authors · 2025-08-06
> > >
> > > Thanks for your response, we're happy to hear that we have resolved most of your concerns and stating your remaining concerns which seem to be about the originality of our technique(s) and availability of the experimental details.
> > >
> > > Technique: We're unsure exactly what you mean by this. If you mean our proof techniques, we believe they are novel, and the results do not appear any where else. The core of our paper is not about introducing a technique, as much as it is about answering the fundamental question of whether there are natural settings where sequential token scaling of LLMs is provably better than parallel scaling. We are the first to show such a setting and prove a tradeoff. (We also show empirically that this tradeoff occurs in LLMs trained with standard techniques)
> > >
> > > Experimental Details: The experimental details you asked about are all included in the paper as stated in our rebuttal, and we included all of our code in our original submission. Let us know if there's anything we're missing, or any other concerns about the experimental details.

---

> > > ### Author Response · Authors · 2025-08-08
> > >
> > > In response to your valuable feedback, and to assess the generalizability of our conclusions to other reasoning tasks, such as mathematical problem solving, we conducted additional experiments on AIME-2024 and posted the results as an official comment. We hope these results address your remaining concerns and encourage you to consider raising your score.

---

### Official Review · Reviewer_Lhpt · 2025-07-02

**Clarity:** 3
**Significance:** 2
**Originality:** 2
**Rating:** 4
**Confidence:** 4

**Summary:**

This paper explores the trade-off between sequential and parallel scaling of inference-time computation for reasoning tasks using Large Language Models (LLMs). It demonstrates theoretically and empirically that sequential scaling—achieved through long chains of thought (CoT)—can offer an exponential advantage over parallel scaling, such as majority voting or best-of-n sampling over multiple short CoTs. The task used is a graph connectivity problem inspired by complexity theory and graph reasoning literature, specifically focusing on structured "bridge graphs" designed to challenge multi-step reasoning.

The theoretical analysis introduces two key contributions:
1.	Bounded-depth transformers with polynomial-length CoT can solve the connectivity problem efficiently.
2.	Aggregating over polynomially many O(1)-length CoTs cannot achieve similar performance.

**Questions:**

- Would hybrid methods (e.g., iterative refinement of CoTs) narrow the gap?
- Do other architectures (e.g., recurrent reasoning models, O3) exhibit similar limitations?

**Ethical Concerns:**

["NO or VERY MINOR ethics concerns only"]

**Final Justification:**

I have read the authors response. I would keep my score as boarderline accept.

**Limitations:**

Yes

**Paper Formatting Concerns:**

No issues.

**Quality:**

3

**Strengths And Weaknesses:**

Strengths
- The empirical section is comprehensive. The bridge graph dataset is carefully constructed to stress-test sequential reasoning abilities. The evaluation includes ablation studies across CoT lengths, architectures (from scratch-trained Mistral variants to frontier models), and aggregation methods.
- Section 5’s exploration of reinforcement learning-induced growth in CoT length supports both theoretical findings and prior observations in frontier models. This strengthens the relevance and applicability of the results to real-world settings.
- The use of synthetic "bridge graphs" allows precise control over the difficulty of reasoning via increasing depth, offering a reproducible benchmark for future multi-step reasoning research.

Weaknesses
- While the theoretical analysis centers around a specific formal reasoning problem (graph connectivity), its generalization to other types of reasoning remains speculative.  For instance, [1, 2, 3] have explored various graph problems and given a good example for this paper to follow, which should be cited in this paper.
- Theoretical guarantees rely on unproven complexity-theoretic assumptions (TC⁰ ≠ L).
- Empirical evaluation uses small transformers (e.g., 4 layers); implications for massive LLMs are underexplored.
- The paper neglects to cite Can Language Models Solve Graph Problems in Natural Language?[2], which demonstrates that the sequential order of graph descriptions (e.g., BFS vs. DFS vs. PageRank-based orderings) significantly impacts LLM performance on graph reasoning tasks.


[1] Can Language Models Solve Graph Problems in Natural Language?
[2] Can Graph Descriptive Order Affect Solving Graph Problems with LLMs?
[3] Talk like a Graph: Encoding Graphs for Large Language Models

---

> ### Author Rebuttal · Authors · 2025-07-31
>
> Thank you for your thoughtful comments, additional references, and questions which suggest interesting future directions. We are glad that you found our empirical section comprehensive and our carefully constructed bridge graphs suitable as a reproducible benchmark for future multi-step reasoning research. Below we address your concerns.
>
> > While the theoretical analysis centers around a specific formal reasoning problem (graph connectivity), its generalization to other types of reasoning remains speculative.
>
> Our goal in this paper is to make a general and fundamental claim about tradeoffs between parallel and sequential scaling. Our understanding of these tradeoffs is at its infancy. Multiple papers in the literature have seemingly contradictory claims: some show large benefits from sequential scaling [1], while other papers claim that parallel scaling alone is sufficient [2].
>
> Our goal is to add clarity to the literature, by providing a controllable and analyzable task in which sequential scaling (theoretically & empirically) cannot be efficiently replaced by parallel scaling. Thus, our results yield the general conclusion that the parallel scaling recipes of papers such as [2, 3] cannot work for all types of problems.
>
> Furthermore, we propose the graph connectivity task because it is very natural and captures multi-step reasoning ability, which seems to be a key aspect of more complex reasoning tasks such as math problems. Reasoning on graphs has been considered in the literature as an ideal abstraction of complex reasoning tasks (see [4,5,6,7] for examples), which also isolates the reasoning ability from memorization. Motivated by that, we have designed the bridge graph connectivity task, to capture key aspects of multi-step reasoning.
>
> To be clear, our claim is not that sequential scaling is always better than parallel scaling for all tasks. Rather, it is that there is a natural class of tasks where it is (empirically and theoretically) better. **We are the first to formally show such a gap for transformers in a natural setting that is a building block for more advanced multi-step reasoning tasks.** This is a conceptual contribution that we believe is helpful for thinking about more complex tasks. We will edit the introduction and abstract to be more clear on this point so as to avoid confusion.
>
> Finally, we have run some evaluations of s1 [1] on GPQA Diamond, and compared sequential and parallel scaling in that setting (we cannot include any plots due to the removal of the option to share a pdf). These experiments show sequential vs. parallel tradeoffs qualitatively similar to our experiments on the graph connectivity task. As argued above, we believe that these experiments are not core to our message and that the paper stands on its own without them. Nevertheless, we will include these experiments in the camera ready paper, since they support our claim that graph connectivity is a helpful benchmark to consider.
>
> > For instance, [1, 2, 3] have explored various graph problems and given a good example for this paper to follow, which should be cited in this paper.
>
> Thank you for providing these references, which provide empirical studies of LLMs on graph reasoning benchmarks. They complement our theoretical and empirical analyses very nicely, and we will cite and discuss them thoroughly in the final version of our paper.
>
> > Theoretical guarantees rely on unproven complexity-theoretic assumptions ($\textrm{TC}^0 \neq \textrm{L}$).
>
> The assumption that $\textrm{TC}^0 \neq \textrm{L}$ is a standard assumption (see [8,9]) commonly held by complexity theorists (similarly to the conjecture that $\textrm{P} \neq \textrm{NP}$). We explicitly mention this assumption whenever we use it, and also have theoretical results (see Theorem 1) that do not rely on this assumption.
>
> > Empirical evaluation uses small transformers (e.g., 4 layers); implications for massive LLMs are underexplored.
>
> We have accompanied our from-scratch trained experiments with small transformers with empirical evaluation on massive LLMs, discussed briefly in section 4.5, with results mostly in figures 1b, 5, and 8 (in the appendix). Specifically, we looked at three separate 32 billion parameter models, and tested the limits of parallel and sequential compute in our setting on these models. We will make sure to edit our abstract to make it more clear that the “large reasoning models” (line 12) we refer to are indeed LLMs. This important component of our results may not have been emphasized enough, so we will add more discussion about the LLM experiments, and emphasize it earlier on.
>
> > The paper neglects to cite Can Language Models Solve Graph Problems in Natural Language?[2], which demonstrates that the sequential order of graph descriptions significantly impacts LLM performance on graph reasoning tasks.
>
> Thanks for the reference! We will cite this paper in the camera-ready.
>
> > Would hybrid methods (e.g., iterative refinement of CoTs) narrow the gap?
>
> This is a very interesting question, which is definitely worthy of future research. With a variety of bottlenecks determining the efficiency of parallel and sequential computation in practice, it would not be surprising at all if hybrid methods could outperform both of the simple strategies discussed in the paper.
> We do conjecture that any hybrid method that involves generating and $O(1)$ number of $O(1)$-length chains of thought for $O(1)$ rounds would fail at our graph reasoning tasks – the proof would rely on similar ideas to what we use for Theorem 1, but we would have to check the details more carefully before claiming such a statement.
>
> > Do other architectures (e.g., recurrent reasoning models, O3) exhibit similar limitations?
>
> The State-Space Model (SSM) architecture was also shown to be in $\textrm{TC}^0$ [10], so our first theorem on theoretical results on limitations of sequential scaling also applies to it. We will add a note of this in the paper, thank you for your suggestion! We can only speculate on the architecture in proprietary models such as O3, but it is possible that it is a transformer, or hybrid between transformer and SSM, in which case these results should also still apply.
>
> Thank you again for your careful reading of the paper and thoughtful suggestions. We hope that we have addressed your questions sufficiently.
>
> [1] Muennighoff, Niklas, et al. "s1: Simple test-time scaling." arXiv preprint arXiv:2501.19393 (2025).
>
> [2] Ma, Wenjie, et al. "Reasoning models can be effective without thinking." arXiv preprint arXiv:2504.09858 (2025).
>
> [3] Brown, Bradley, et al. "Large language monkeys: Scaling inference compute with repeated sampling." arXiv preprint arXiv:2407.21787 (2024).
>
> [4] Abbe, Emmanuel, et al. "How far can transformers reason? the globality barrier and inductive scratchpad." Advances in Neural Information Processing Systems 37 (2024): 27850-27895.
>
> [5] Sanford, Clayton, et al. "Understanding transformer reasoning capabilities via graph algorithms." Advances in Neural Information Processing Systems 37 (2024): 78320-78370.
>
> [6] Xu, Keyulu, et al. "What can neural networks reason about?." arXiv preprint arXiv:1905.13211 (2019).
>
> [7] Kim, Juno, et al. "Metastable dynamics of chain-of-thought reasoning: Provable benefits of search, rl and distillation." arXiv preprint arXiv:2502.01694 (2025).
>
> [8] Feng, Guhao, et al. "Towards revealing the mystery behind chain of thought: a theoretical perspective." Advances in Neural Information Processing Systems 36 (2023): 70757-70798.
>
> [9] Merrill, William, and Ashish Sabharwal. "The expressive power of transformers with chain of thought." arXiv preprint arXiv:2310.07923 (2023).
>
> [10] William Merrill, Jackson Petty, and Ashish Sabharwal. “The Illusion of State in State-Space Models”. ICML 2024.

---

> > ### Author Response · Authors · 2025-08-08
> >
> > In response to your valuable feedback, and to assess the generalizability of our results to other types of reasoning, we conducted additional experiments on AIME-2024 and posted the results as an official comment. We hope these results address your remaining concerns and encourage you to consider raising your score.

---

> ### Author Response · Authors · 2025-08-05
>
> We hope our review addressed your concerns to your satisfaction, sufficiently for you to consider raising your score! We are happy to hear any remaining concerns, confusions, or feedback about the paper.

---

### Author Response · Authors · 2025-08-08
**AIME-2024 Experiments: Further Evidence for Generalizability**

In response to the reviewers’ feedback and to clarify the significance of our fundamental study and its generalizability to more complex reasoning tasks, we conducted additional experiments with the s1‑32B model [1] on AIME‑2024. For parallel scaling, we sample with temperature 1.0 and aggregate by majority vote over final answers. For sequential scaling, we limit the model’s thinking‑token budget and force a final answer once the limit is reached. In the ‘wait’ variant [1], we ignore the model’s first output and append the ‘wait’ token to induce further reasoning before the final answer. The experiments used ≈24 H200 GPU‑hours. Since PDF attachments are not supported here, we present the results in the table below.

| Parallel (maj@k) / Sequential (token) | 500  | 1k   | 2k   | 4k   | 8k   | wait |
|-------------------------------------------------------------|------|------|------|------|------|--------------|
| 1                                                           | 0.067 | 0.133 | 0.200 | 0.333 | 0.300 | 0.433        |
| 2                                                           | 0.067 | 0.133 | 0.200 | 0.333 | 0.300 | 0.433        |
| 4                                                           | 0.100 | 0.133 | 0.233 | 0.433 | 0.433 | 0.500        |
| 8                                                           | 0.067 | 0.133 | 0.300 | 0.433 | 0.467 | 0.533        |
| 16                                                          | 0.067 | 0.167 | 0.333 | 0.433 | 0.500 | 0.567        |
| 32                                                          | 0.067 | 0.133 | 0.400 | 0.433 | 0.533 | 0.600        |
| 64                                                          | 0.067 | 0.133 | 0.400 | 0.433 | 0.533 | 0.567        |
| Avg. #Thinking Tokens                                        | 500  | 1000  | 2000  | 3998  | 5092  | 5522         |



The results show that sequential scaling can not be efficiently replaced by parallel scaling for this mathematical task, supporting the generalizability of our findings to real-world scenarios. While quantifying the exact trade-off between them for complex mathematical problems such as this is beyond the scope of our fundamental study, we observe that the results confirm our conclusion that sequential scaling is necessary, and challenge claims from other works [2] that it can be entirely replaced by parallel scaling.


To further examine this observation, we break down the results by individual question (out of the 30 total). The table below reports the number of correct responses out of 64 attempts for each question.

| Sequential (token budget) / Question ID |  1 |  2 |  3 |  4 |  5 |  6 |  7 |  8 |  9 | 10 | 11 | 12 | 13 | 14 | 15 | 16 | 17 | 18 | 19 | 20 | 21 | 22 | 23 | 24 | 25 | 26 | 27 | 28 | 29 | 30 |
|---------------------------|----|----|----|----|----|----|----|----|----|----|----|----|----|----|----|----|----|----|----|----|----|----|----|----|----|----|----|----|----|----|
| 500                       |  0 |  0 |  0 |  0 |  1 |  0 |  0 | 36 | 12 |  0 |  0 |  0 | 13 |  0 |  0 |  0 |  0 |  2 |  3 |  0 |  0 |  0 |  0 |  0 |  0 |  0 |  0 |  0 |  0 |  0 |
| 1k                        |  0 |  1 |  0 |  0 |  6 |  0 |  0 | 55 | 36 |  4 |  0 |  4 | 48 |  0 |  0 |  2 |  0 |  1 |  1 |  0 |  0 |  1 |  0 | 27 |  0 |  0 |  0 |  0 |  0 |  0 |
| 2k                        | 56 |  1 |  0 |  0 | 20 |  0 |  0 | 63 | 23 | 55 |  0 | 29 | 47 |  0 |  1 |  9 |  1 |  1 |  4 | 12 |  0 |  0 | 12 | 34 | 57 |  0 | 15 |  0 |  0 |  0 |
| 4k                        | 58 |  1 |  0 |  0 | 16 |  0 |  2 | 63 | 22 | 60 |  2 | 43 | 58 |  1 |  4 | 27 |  0 |  3 |  3 | 31 |  0 |  0 | 36 | 50 | 62 |  0 | 33 |  5 |  0 |  0 |
| 8k                        | 62 |  2 |  0 |  0 |  5 |  2 | 24 | 64 | 28 | 63 |  1 | 49 | 58 |  9 | 16 | 40 |  2 |  2 | 16 | 33 |  2 |  0 | 29 | 51 | 63 |  2 | 36 |  8 |  0 |  0 |
| wait                      | 61 |  0 |  0 |  0 |  5 |  0 | 35 | 64 | 22 | 62 |  6 | 48 | 60 | 12 | 19 | 44 |  7 |  6 | 19 | 34 |  0 |  0 | 31 | 50 | 60 |  2 | 31 |  7 |  0 |  0 |



One notable takeaway is that there exist questions (e.g. 1, 10, 25) for which parallel scaling at low token budgets will never succeed, but a single shot with a higher token budget is very likely to succeed. This shows the necessity of sequential scaling for achieving the highest possible score on this task. We hope this further clarifies the generalizability of the insights offered by our work.

[1] Muennighoff, Niklas, et al. "s1: Simple test-time scaling." arXiv preprint arXiv:2501.19393 (2025).

[2] Ma, Wenjie, et al. "Reasoning models can be effective without thinking." arXiv preprint arXiv:2504.09858 (2025).

---

### Note · Authors · 2025-08-13

We thank the reviewers for their thoughtful comments and helpful suggestions. We appreciate that our theoretical framework was described as “novel” (GJ4H), with “technical depth” (WjKT) and “solid theoretical contributions” (GJ4H). We are also pleased that our “carefully constructed” (Lhpt) task was regarded as an “ideal testbed” (2f9M) for studying the trade-off between sequential and parallel scaling, offering a “challenging” (2f9M) and “reproducible benchmark for future multi-step reasoning research” (Lhpt).

We are glad the empirical section was found “comprehensive” (Lhpt), with “thorough investigation” (WjKT) and “sufficient empirical validation” of our theoretical predictions (GJ4H, 2f9M). We also appreciate the recognition of our work’s practical relevance, including “solid theoretical and empirical contributions to understanding inference-time scaling, which is an important problem in current LLM research” (GJ4H), as well as the reinforcement learning section, which “contributes to the novelty of the work” (WjKT) and “strengthens the relevance and applicability of the results to real-world settings” (Lhpt).

During the discussion period, we addressed all questions regarding our experimental details and theoretical assumptions, and clarified the significance of our study, being the first to formally establish a gap between sequential and parallel scaling for transformers in a natural setting that serves as a building block for more advanced multi-step reasoning tasks.

Furthermore, in response to constructive feedback on the generalization of our results to other reasoning tasks, we conducted an analysis with the s1-32B model on the AIME-2024 dataset. The results, posted as an official comment, further support our theoretical findings and demonstrate the generalizability of our claims to more complex reasoning tasks. We believe this reinforces the significance of our fundamental study in illuminating the landscape of inference-time scaling.

---

### Decision · Program_Chairs · 2025-09-17

**Decision:**

Accept (poster)

**Comment:**

This paper studies the question of sequential vs parallel inference time scaling for a graph connectivity task. At a high level, the paper shows that while sequentially scaling with one polynomial-length CoT can solve the connectivity problem, parallel scaling by aggregating over polynomially-many O(1) CoT fails. The reviews are mostly positive. The reviewers agree that this is an fundamental problem to study given the importance of inference time scaling. While there were minor concerns about the scope of the theoretical analysis and novelty, I think it is useful for the broader community to know about these results. I recommend acceptance of the paper.